# A study on the promotion effect of government guidance on the construction of a national unified market logistics channel

**Guangsheng Zhang**[1], **Junqian Xu**[2]*, **Yanling Wang**[1]

**1** College of Business Administration, Shandong Management University, Jinan City, China, **2** School of Accountancy, Wuxi Taihu University, Wuxi City, China

* jqxu2000@126.com

**Data Availability Statement:** No data was used for the research on the topic of game theory model in this manuscript.

**Funding:** Funding 1: Funding Name: National Social Science Foundation of China. Award

## Abstract

Logistics channel is the lifeblood to ensure that logistics serves the circulation inside and outside the region, and to realize regional economic integration, it greatly contributes to the implementation of the national unified market strategy. As the government plays an important role in the construction of logistics channels, this paper further clarifies the effect of government participation and support policies by defining the role and functions of the government in the construction of logistics channels. Based on the evolutionary game theory, the paper reveals the equilibrium conditions of logistics channel construction under the market mechanism and government guidance under the assumption of bounded rationality. We construct an evolutionary game model among participating stakeholders, then study the evolutionary stability strategy of logistics channel participation behavior using the stability theorem for the model's differential equations. In order to explore the dynamic evolution process of both parties' choices under the two modes, we investigates the influence of the initial intention, cooperative income, cost proportion, penalty coefficient and construction cost of participating enterprises on both parties' strategic decisions under the market mechanism and government guidance modes through numerical simulation. We find that: (1) under the market mechanism and government guidance modes, there is a game equilibrium in the participation behavior of logistics enterprises in the national unified market, and that the conditions for realizing the equilibrium of cooperation among stakeholders under the guidance of the government are easier to meet; (2) The initial intentions of the two players in the game along the logistics channel influence each other, and government participation can change the effects of cooperative income, the penalty coefficient and construction cost on the system game strategy, which has a positive effect on the channel construction; (3) At the same time the simulation shows that the government's promotion effect has certain limitations, and the government should provide reasonable guidance to prevent enterprises from hindering the healthy development of logistics channels. This study provides a theoretical reference for the government and logistics enterprises, especially relying on logistics channels to support the regional coordination of national unified market development.

Number: 21BJY227 Recipient: Guangsheng Zhang Funding 2: Funding Name: Major Project of Philosophy and Social Science Research in Colleges and Universities of Jiangsu Province Award Number: 2023SJZD058 Recipient: Junqian Xu Funding 3: Funding Name: Wuxi philosophy and social science bidding project Award Number: WXSK23-C-14 Recipient: Junqian Xu Funding 4: Funding Name: the PhD Research Initiation Fund of Shandong University of Management Award Number: None Recipient: Guangsheng Zhang The funders had no role in study design, data collection and analysis, decision to publish, or preparation of the manuscript.

**Competing interests:** The authors have declared that no competing interests exist.

# 1. Introduction

In recent years, the complexity, severity and uncertainty of the global economic development environment have increased. Building a unified national market is conducive to accelerating the construction of a unified, open and competitive modern market system and realizing high-quality economic development. Historically, developed countries have established their leading position in industrial software, components and high-end machinery manufacturing industries by virtue of their strong unified market competition, fair market supervision and high-standard Unicom technology. For example, the United States, as the most economically developed country in the world, unifies its capital market and provides a good environment for the development of the semiconductor industry. From the 1970s to 1980s, the U.S. economy fell into 'stagflation' due to the imbalance between supply and demand. During this period, because the semiconductor industry was still in its infancy, it was difficult for small and medium-sized enterprises to obtain bank credit and bond financing. Against this background, the U.S. government strengthened its dominant position in the semiconductor industry by setting up venture capital and technology alliances. In March 2021, the Chinese government proposed to accelerate the establishment of a "dual circulation" development pattern in which the domestic economic cycle plays a leading role while the international economic cycle remains its extension and supplement during the '14th Five-Year Plan' period. The key to building a new development pattern lies in the unimpeded economic circulation, which requires the combination of various factors of production to be organically linked and circulated in all aspects of production, distribution, circulation and consumption. Therefore, it is urgent to speed up the construction of a unified national market, break the local protection and market segmentation, break through the key constraints that restrict the economic process, and promote the smooth flow of commodity factor resources at a larger scale. In April 2022, Chinese government set out *the Opinions of the State Council*, *the Central Committee of the Communist Party of China on Accelerating the Construction of a National Unified Market*, further extending overall arrangements for accelerating the construction of a national unified market, and breaking the market segmentation caused by local protection and regional barriers by establishing unified market rules, elements, resources and supervision and interconnected logistics. Modern logistics is the basic skeleton of a unified national market. In order to build a modern logistics system, we must take the specialization and cooperation of regional logistics organizations as a strategic task, and improve the system and its mechanisms to give full play to the enthusiasm of the government and enterprises [1, 2].

A logistics channel refers to the main arteries and framework of material flow formed by a comprehensive transportation network, which connects production and consumption, connects logistics nodes in series, and connects the market. It is an important carrier of the trade channel and national economic corridor [3], and it is an important support to ensure that logistics serves the circulation inside and outside the region and realizes regional economic integration [4, 5]. Promoting the construction of a logistics channel aims at improving the carrying capacity and span of logistics, efficiently leading the agglomeration, intensive allocation, integrated optimization of logistics resources, and strengthening regional logistics interconnection [6, 7]. At present, the most difficult and urgent problem to be solved is to lower the barriers of division across administrative regions and industries [8, 9], while a smooth logistics channel is the prerequisite of a sound and functional trans-regional logistics network. A smooth logistics channel is also an important entry point to lowering the current segmentation barriers, and an effective path to promote the effective integration of logistics resources, reduce costs and increase efficiency [10]. In this context, the construction of logistics channel not only affects the competition and profit changes among enterprises in the market, but also inevitably

affects the strategic decision of enterprises. Relevant subjects often enhance competitiveness and expand market share through cooperation in different ways. Most of the related studies use traditional game theory to analyze the cooperative relationship among various stakeholders, and have achieved corresponding results in the aspects of strategic decision-making [11, 12], enterprise investment [13] and supply chain cooperation [14]. At present, the research on the decision-making of logistics channel construction mainly focuses on the cooperation of supply chain enterprises, and all studies are assumed to be rational participants. Few studies consider the combination of logistics channel with bounded rationality, and few studies incorporate the government into the decision-making framework of logistics channel.

The excellent logistics channel system cannot be formed internally by the market mechanism and the economic system under its control. Therefore, the development of regional logistics channels has an internal dependence on the government departments. Even the more developed western countries have also experienced the construction process of government-led regional logistics system construction. The government plays an important role in the construction of logistics channels, including formulating strategies and leading the direction, shaping the environment, coordinating the economic and social interests among the subjects, and providing infrastructure and generic technology for the subjects to carry out applied research. However, the current government participation policy in the construction of logistics channel has failed to achieve the ideal effect, which has triggered the debate on the government function and behavior in the theoretical circle. Therefore, it is of great practical significance to clarify the roles and functions of the government in the construction of logistics channels, and further clarify the effect of government participation and support policies for guiding the construction of China's logistics industry and the implementation of high-quality development strategy. Under the national unified market strategy, the government is intended to play a direct or indirect guiding role in the interaction between enterprises along the logistics channel. For the government, how to formulate reasonable and effective support policies, encourage enterprises to participate in the construction of logistics channels, and promote logistics enterprises to improve the efficiency of cross-regional cooperation, has become an urgent problem to be solved. However, existing research largely neglects the role of the government and its interaction with the cooperation decision of the logistics channel. In this regard, it is an urgent practical problem for the government to formulate appropriate guiding policies to encourage enterprises to participate in the construction of logistics channels, and for enterprises to make decisions to achieve a win-win situation under such policy guidance. Evolutionary game theory is different from traditional game theory, evolutionary game theory does not require participants to be completely rational, nor does it require complete information conditions, which can more truly reflect the diversity and complexity of the logistics channel actors, and can provide theoretical basis for macro-control group behavior. Therefore, exploring the evolution of the game behavior of logistics enterprises participating in logistics channels under the guidance of the government is helpful to overcome the failure of market resource allocation and realize benefit sharing, which has certain theoretical value and practical significance. This paper focuses on four key issues as follows:

1. What are the conditions for the construction of logistics channels under the market mechanism and government guidance under the assumption of bounded rationality? Is there a stability strategy for the evolutionary game model among participating stakeholders?

2. Under the two intervention and non-intervention modes, what is the relationship between the equilibrium convergence of each player and its initial strategy in the evolution process of regional logistics internal and external participating enterprises? How does the initial strategy affect the stability of the strategy?

3. Can the reward and punishment mechanism under the guidance of the government change the final stable state of the logistics channel evolution system? Is there an effective interval between the two sides' game strategies?

4. What are the effects of cooperative income, cost ratio, penalty coefficient and construction cost of participating enterprises on the strategic decision-making of both parties under the market mechanism and government guidance mode?

The main contributions of this paper are as follows: firstly, unlike traditional research, this paper starts from the mechanism of the government guiding the construction of logistics channels under the limited rational conditions, a model is constructed whereby under the unified national market, the government is taken as the participating department, and logistics enterprises are encouraged to participate in playing in an evolutionary game model of logistics channel construction through rewards and punishments, the stability theorem of the differential equation is used to explore the evolutionary stability strategy of the participation behavior of the logistics channel; Secondly, we compares the differences of the participation behavior of the logistics channel between market mechanism and government guidance, analyzed the difficulty of realizing and maintaining the system stability under these two modes. The results show that the participation behavior game strategy of logistics channel under government guidance is easier to achieve evolution and stable equilibrium; Thirdly, the influence effect of the initial intention, cooperation benefit, cost proportion, penalty coefficient and construction cost on the strategic decision of both sides under the two modes are explored, we found that government participation can change the effect of cooperation income, penalty coefficient and construction cost on the system game strategy. These results can well complement the existing studies and provide some novel managerial insights for the operations management of the national unified market logistics channel.

The remainder of this paper is organized as follows. In Section 2 we summarize the related literature. Section 3 develops decision models under two participants. Section 4 conducts the simulation and sensitivity analysis. Section 5 provides conclusions, management insights, and future directions for research.

## 2. Literature review

The research related to this paper can be divided into two types: firstly there is the literature on the construction of logistics channels with government participation and secondly there is the literature on the application of game theory in the coordination of stakeholders in logistics channels, focusing on the influence effect of participants' behavior on regional logistics cooperation.

### 2.1 Research on the construction of logistics channels with government participation

In the construction of the modern industrial chain and supply chain under the background of economic globalization, the logistics industry is playing an increasingly important role [15–17], the logistics service organization relying on transportation channels and logistics hub shows an obvious trend of high efficiency and low cost [18, 19]. In recent years, some scholars have studied the construction of logistics channels, for example, Wang and Chou (2020) systematically designed the logistics channel, logistics nodes and logistics formats of "Belt and Road", and put forward the overall structure and construction plan of the "One Belt And One Road" logistics infrastructure promotion policy [20]. Zhang and Xiang (2022) used three-stage DEA and random frontier analysis methods to evaluate the railway logistics efficiency of the

Belt and Road countries, including technical efficiency, pure technical efficiency and scale efficiency, and pointed out that the system mechanism and interest pattern become important factors restricting the construction of logistics network system and logistics channel [21]. The above research puts forward the importance of logistics channel to regional economic development, and discusses the evaluation method of regional logistics channel development. At the same time, for the construction of logistics channels, government participation in the rational layout and construction of logistics hubs [22, 23], connecting different countries, industries and various links, mining the industrial value-added value of the countries along the corridor, realizing the cost reduction and efficiency increase of logistics itself and related industries [24, 25], all have an increasingly important role. Therefore, some studies focus on the mechanism of government participation strategy on the construction of logistics channels [25, 26]. For example, Zhang et al.(2019)introduced government subsidy policies and carbon emission trading policies to guide cold chain logistics enterprises to carry out energy saving and emission reduction transformation of cold storage, and applied this method to the decision-making of regional cold chain logistics system in Wuhan, Hubei Province [27]. Hu et al. (2022) established a logistics cooperation model of cross-border two-way supply chain under the benefit sharing and cost sharing policy, and put forward strategies for effective cooperation and coordination of supply chain and reducing the impact of tariffs on the global trading system [28]. Babagolzadeh et al. (2022) analyzed the structure and decision of regional distribution network under different government subsidy schemes, and the results showed that the introduction of subsidies can effectively reduce the total logistics cost [29]. Different from the above literature, this paper starts from the mechanism of government guidance on the construction of logistics channel, explores the dilemma of "market failure" under the market mechanism and the optimization of innovation mode under government guidance, so as to provide scientific suggestions for enterprises participating in the construction of regional logistics channel.

## 2.2 Logistics channel stakeholders game coordination related research

High-quality logistics service capabilities often benefit from the joint efforts of stakeholders [30, 31], there are still many cooperation difficulties in the collaborative construction of cross-regional logistics channels of local governments, leading to the increase of transaction costs of all parties and the insignificant synergistic efficiency and benefits [32]. Game theory is able to consider the predictive and actual behavior of logistics channel stakeholders and study their optimization strategies [33, 34]. Game theory is able to consider the predictive and actual behavior of logistics channel stakeholders and study their optimization strategies [33, 34]. Through the game model, the expected benefits and losses of the logistics channel participants can be analyzed, and the benefits and payment costs obtained by the interest subjects when choosing different strategies can also be investigated. As long as the cooperation benefits of all parties are greater than the cost paid for, the cooperation risks can be reduced, the cooperation willingness can be enhanced and their participation can be improved. For example, Yea et al. (2018) used the method of cooperative game to study the strategic alliance of horizontal cooperation in the logistics and transportation industry [35]. The research concluded that by establishing the strategic alliance, operators can gain additional benefits from resource sharing and efficient resource utilization. Zhang et al. (2019) studied the existence of Nash equilibrium by establishing the guno game and Stackelberg game models [36]. Deng et al. (2022) developed A three-player evolutionary game model (TEGM) to analyze the interaction between freight companies, freight shippers and logistics platforms in the regional logistics channel [37], studied the asymptotic equilibrium and evolutionary stability strategy of tripartite game. In the context of metro-integrated logistics systems (MILS), Ma et al. (2022) analyzed the strategic

interaction between subway and logistics operators and the resulting whole system influence, and put forward the realization path of the transportation efficiency of the modern urban logistics channel [38]. To sum up, many scholars have adopted the game method to explore the problems related to logistics channels, but mainly for the theme of regional logistics alliance and logistics cooperation. Although some scholars take into account the influence of the government reward and punishment mechanism on the logistics channel, they only build the evolutionary game model between the government and enterprises in essence, and do not compare the market mechanism with the behavior decision of the participants in the logistics channel under the guidance of the government, which is difficult to reflect the difficulty of the logistics channel system to reach equilibrium under the two modes.

In summary, this paper assumes that the individuals participating in the logistics channel are bounded rational subjects, analyzes the equilibrium stability of the stakeholders' behavior based on evolutionary game theory, and explores the promotion effect and optimal path of government guidance on the logistics channel construction in five dimensions: willingness to participate, construction cost, cooperation income, sharing ratio and punishment coefficient, and explores the effective countermeasures for the "market failure" dilemma.

## 3. Basic assumptions and model building

The construction of the logistics channel of the national unified market is characterized by openness and diversity. There are four types of participants in the construction of the logistics channel, namely, the logistics enterprises in the region seeking foreign cooperation, the enterprises outside the region along the channel, the government organization and the intermediary organizations [39, 40]. Among them, logistics enterprises are the main body undertaking channel construction, promoting cross-regional logistics cooperation, resource sharing and regional linkage development, so as to create cross-regional logistics resource development and realize economic growth under benefit sharing. Logistics enterprises are the main body of government policy innovation, responsible for the incentive structure of rewards and punishments, management and supervision. The intermediary is the main body of service guarantee, providing the platform, information and management services for all kinds of organizations through professional knowledge and technology, and charging a fee. Cross-regional enterprises are selected as the two main players in the process of logistics channel construction, and the evolutionary game models of logistics channel construction guided by market mechanism and government are constructed respectively.

### 3.1 Model assumptions

Under the unified national market, enterprises along the logistics channel seek foreign cooperation, and there are two strategic choices in the evolutionary game model: "active cooperation" and 'passive cooperation'. Of these, 'active cooperation' refers to the cooperation between the two parties in accordance with the contract, the reasonable sharing of resources and information, the joint risk of cooperation, the sharing of cooperation results, and the realization of logistics resources sharing and complementary advantages; 'passive cooperation' refers to the cooperative speculation of participating enterprises, that is, on the one hand, enterprises protect their market position and have private resources and information in the process of cooperation, on the other hand, they have the mentality of "free-riding" and enjoy the fruits of cooperation with less investment. Here are a series of assumptions for the game process of both enterprises:

Hypothesis 1: The players are two cross-regional logistics enterprise groups. They are bounded rational players with independent learning ability and independent decision-making power.

Hypothesis 2: There is information asymmetry in the game between the two sides. Logistics enterprises do not know the real business situation of the other party, that is, for one party in the game, the information and behavior of the other party are incomplete.

Hypothesis 3: The two sides play a non-zero-sum game. The purpose of logistics channel construction is to achieve the ultimate win-win situation in a national unified market, rather than zero-sum competition.

Hypothesis 4: The two sides play repeated games. The construction of the national unified market logistics channel is a dynamic process, that is, the two sides constantly adjust the game strategy through repeated trial and error, learning, and finally reach a balanced state.

## 3.2 Research on the construction conditions of national unified market logistics channel under the market mechanism

**(1) Payment function.**   In our model we look at play between players inside a region and players outside. The establishment and maintenance of the cross-regional logistics channel in the unified national market needs to invest in certain production factors, including land, labor, capital, technology and information and management. It is assumed that the revenue of the cross-regional logistics channel is only positively related to the input cost, and has nothing to do with other factors such as degree of contribution and participation enthusiasm. We set the cost of logistics enterprises inside and outside the region to participate in the construction of logistics channels as $C_1$, and the cost paid to the intermediary organization is $C_2$. Among them, the proportion of enterprises inside and outside the region is $p$, $(1 - p)$ respectively, and $0 < p < 1$. Under the unified national market, no matter whether the logistics enterprises inside and outside the region choose the strategy of 'active cooperation' to build logistics channels, the cooperation benefits obtained by the basic cooperation level within and without the region are earnings $Q_1$ of logistics enterprises inside the region $I$, and income $Q_2$ of logistics enterprises outside the region $O$. At this time, there are the following types of benefits: ① When the enterprises inside and outside the region do not actively participate in the channel construction, the benefits obtained by both parties are $Q_1$ and $Q_2$; When both of them actively cooperate to build logistics channels, they can obtain excess profits $S$, with the proportion of construction investment as the distribution standard, the income of logistics enterprises inside the region $pS$, the corresponding income of logistics enterprises outside the region is $(1 - p)S$, the benefits of both also have positive incentive effect of positive cooperation, that is, the more frequent the cooperation, the better the positive effect, $\delta$ indicates the incentive effect; When the two parties choose to actively cooperate in the early stage of logistics channel construction, but one party continues to participate in the cooperation and the other party gives up the cooperation during the construction process, the one that gives up the cooperation will get extra income because of free riding.$R_i (i = 1, 2)$, but needs to pay a compensation fine to the partner. $\lambda$ is set as the penalty coefficient, and the compensation amount is $\lambda S$.

Under the market mechanism, in the game stage of logistics channel construction in the national unified market, it is assumed that the positive cooperation probability of logistics enterprises in the region is $x$, the probability of active cooperation among logistics enterprises outside the region is $y$. Logistics channel promotion game $n$ times, the evolutionary game payment matrix is shown in Table 1.

**Table 1. Game payment matrix of logistics channel enterprises in market mode.**

| | | Regional external logistics enterprises | |
|---|---|---|---|
| | | Active cooperation $y$ | Passive cooperation $(1-y)$ |
| Regional internal logistics enterprises | Active cooperation $x$ | $Q_1 + p[(1+\delta)^{n-1} S - C_1 - C_2]$; $Q_2 + (1-p)[(1+\delta)^{n-1} S - C_1 - C_2]$ | $Q_1 + \lambda S - p[C_1 + C_2]$; $Q_2 + R_2 - \lambda S$ |
| | Passive cooperation $(1-x)$ | $Q_1 + R_1 - \lambda S$; $Q_2 + \lambda S - (1-p)[C_1 + C_2]$ | $Q_1$; $Q_2$ |

**(2) Stabilization strategy.** Under bounded rationality, the logistics channel participation behavior is a complex and dynamic process, and before the final equilibrium is achieved, the game player strategy is in a state of constant adjustment. In the group composed of boundedly rational players, the dynamic changes of the expected benefits of 'active cooperation' and 'passive cooperation' prompt the participants in regional logistics channel construction to adjust their behaviors in real time, and began to learn, imitate or introduce advanced behavior patterns. The dominant strategy is gradually adopted by more players. When $t$ changes the cooperation probability of participants over time, the evolutionary game of logistics channel construction obviously supports the replication dynamic theory. Therefore, the probability of positive cooperation between logistics enterprises inside and outside the region $x$ and $y$ is a function of $t$, or $x(t)$ and $y(t)$, the dynamic differential equation of both parties' adjustment strategy $dx(t)/dt$, $dy(t)/dt$ has a positive correlation with the difference between the expected return and the average return. The dynamic evolution process of behavior strategy is analyzed based on dynamic replication differential equation analysis, according to the stability principle of differential equations, the stable point of replication dynamic equation needs to realize $F(x) = 0$, $f(x) = dF(x)/dx < 0$. This section uses the above principles to analyze the stability of participants in logistics channel construction and their evolution paths.

Suppose that in the process of evolutionary game, regional logistics enterprises ($I$) get expected benefits from implementing the 'active cooperation' strategy $U_{I1}$, the implementation of the 'passive cooperation' strategy is expected to yield $U_{I2}$, average expected income $\bar{U}_I$, according to the income matrix, we get:

$$U_{I1} = y[Q_1 + p((1+\delta)^{n-1} S - C_1 - C_2)] + (1-y)[Q_1 + \lambda S - p(C_1 + C_2)] \qquad (1)$$

$$U_{I2} = y[Q_1 + R_1 - \lambda S] + (1-y)Q_1 \qquad (2)$$

$$\bar{U}_I = x U_{I1} + (1-x) U_{I2} \qquad (3)$$

Therefore, the dynamic replication differential equation of logistics enterprises inside the region is as follows:

$$F(x) = dx(t)/dt = x(1-x)(\bar{U}_I - U_{I1}) =$$
$$x(1-x)[y(pS(1+\delta)^{n-1} - R_1) + \lambda S - p(C_1 + C_2)] \qquad (4)$$

Similarly, logistics enterprises outside the region ($O$) get expected benefits from implementing 'active cooperation' strategy $U_{O1}$, the implementation of the 'passive cooperation' strategy

is expected to yield $U_{O2}$, average expected income $\bar{U}_O$, according to the income matrix, we get:

$$U_{O1} = x[Q_2 + (1-p)((1+\delta)^{n-1}S - C_1 - C_2)] + (1-x)[Q_2 + \lambda S - (1-p)(C_1 + C_2)] \quad (5)$$

$$U_{O2} = x[Q_2 + R_2 - \lambda S] + (1-x)Q_2 \quad (6)$$

$$\bar{U}_O = yU_{O1} + (1-y)U_{O2} \quad (7)$$

Therefore, the replication dynamic differential equation of logistics enterprises inside the region is as follows:

$$\begin{aligned} F(y) = dy(t)/dt = y(1-y)(\bar{U}_O - U_{O1}) = \\ y(1-y)[x((1-p)S(1+\delta)^{n-1} - R_2) + \lambda S - (1-p)(C_1 + C_2)] \end{aligned} \quad (8)$$

In the evolutionary game dynamic system, let $F(x) = 0$, $F(y) = 0$, five pure strategy Nash equilibrium points we get are: $O_1(0, 0)$, $O_2(0, 1)$, $O_3(1, 0)$, $O_4(1, 1)$, $O_5\left(\frac{(1-p)(C_1+C_2)-\lambda S}{(1-p)S(1+\delta)^{n-1}-R_2}, \frac{p(C_1+C_2)-\lambda S}{pS(1+\delta)^{n-1}-R_1}\right)$. According to the Lyapunov stability theory, the main conditions of logistics channel construction are found, and the key factor of the Pareto optimal equilibrium $O_4(1, 1)$ is achieved by using Jacobian matrix. At this time, participants in the Jacobian matrix $J$ are as follows:

$$J = \begin{bmatrix} (1-2x)[y(pS(1+\delta)^{n-1} - R_1) + \\ \lambda S - p(C_1 + C_2)] & x(1-x)(pS(1+\delta)^{n-1} - R_1) \\ \\ y(1-y)((1-p)S(1+\delta)^{n-1} - R_2) & (1-2y)[x((1-p)S(1+\delta)^{n-1} - R_2) + \lambda S \\ & -(1-p)(C_1 + C_2)] \end{bmatrix}$$

Substituting balance point $O_4(1, 1)$ into the Jacobian matrix of the system to obtain the Jacobian matrix of Nash equilibrium point, the eigenvalue matrix of $J_1$ is as follows:

$$J_1 = \begin{bmatrix} -pS(1+\delta)^{n-1} + R_1 - \\ \lambda S + p(C_1 + C_2) & 0 \\ \\ 0 & -(1-p)S(1+\delta)^{n-1} + R_2 - \lambda S \\ & +(1-p)(C_1 + C_2) \end{bmatrix}$$

When all the eigenvalues of the Jacobian matrix are negative, its equilibrium point is the stable point of game system evolution, so it can be known that it satisfies the conditions $R_1 - \lambda S < pS(1+\delta)^{n-1} - p(C_1 + C_2)$, and $R_2 - \lambda S < (1-p)S(1+\delta)^{n-1} - (1-p)(C_1 + C_2)$, $O_4(1, 1)$ is an evolutionary stable point. According to the above conditions, it is easy to know that an excess return $S$ of the positive cooperation of the participants in the construction of logistics channels, incentive effect $\delta$, cooperation frequency $n$ and penalty coefficient $\lambda$, are all positively correlated with the realization of the positive cooperation strategy between the two, and the construction cost of logistics channel $C_1$, intermediary organization cost $C_2$, and other additional benefits $R_i$ are all negatively correlated with the realization of the positive cooperation strategy between the two sides. $R_1 - \lambda S < pS(1+\delta)^{n-1} - p(C_1 + C_2)$ means that logistics enterprises inside the region can get higher profits by choosing the active cooperation strategy, whereby $R_1 - \lambda S$ indicates the net income of logistics enterprises inside the region when they give up cooperation and $pS(1+\delta)^{n-1} - p(C_1 + C_2)$ indicates the net income of logistics

enterprises inside the region when they actively cooperate. Similarly, only when logistics enterprises outside the region actively cooperate to obtain net income $(1 - p)S(1 + \delta)^{n-1} - (1 - p)$ $(C_1 + C_2)$ greater than the net income of abandonment cooperation. $R_2 - \lambda S$, do the two choose active cooperation strategy. Therefore, with a national unified market, the construction of a logistics channel involves many management departments, and is difficult to coordinate. Despite these gains from cooperation, regional blockades and industry monopolies still exist. Compartmentalized, self-contained inter-regional and inter-departmental interest patterns become a challenge to effectively integrating logistics resources and operation, which seriously impedes the improvement of logistics channel functions and efficiency. At this time, if the free-riding benefits of the construction participants along the logistics channel are high, or the cooperation cost of enterprises, especially small and medium-sized enterprises, is too high, the intermediary cost is expensive and the cooperation incentive effect is insufficient, the stability conditions of the above evolutionary game system are difficult to achieve, and the Nash equilibrium of the regional logistics channel cannot be achieved. In such conditions government participation in the construction and coordination of inter-regional and inter-departmental mechanisms can improve outcomes. The government can add value by guiding the construction of cooperative game models of logistics channels, strengthen the coordination and guidance of circulation functions, supervise and inspect, and provide timely solutions to major problems in the process of promotion.

## 3.3 Research on the construction mode of national unified market logistics channel under the guidance of the government

**(1) Payment function.**   According to section 2.2, under the unified national market, the government has a role as the guiding department to encourage logistics enterprises to participate in the construction of logistics channels through rewards and punishments. At this time, the income types from participation can be designed to incentivize participation as follows: ① When the enterprises along the logistics channel do not choose active strategies, the government will not implement rewards or punishments; ② When the logistics enterprises actively participate in the channel construction, the government will give certain funds as business incentives, and the proportion of the reward amount to the total cost is $\varepsilon$; ③ When the logistics channel is constructed, if one party participates in the cooperation and the other party gives up the cooperation, the positive cooperation reward ratio will still be $\varepsilon$, the amount of punishment imposed by the government on the party giving up cooperation is $Z$.

In the evolutionary game guided by the government, it is assumed that the positive cooperation probability of logistics enterprises inside the region is $\alpha$, the probability of active cooperation among logistics enterprises outside the region is $\beta$. Play game of logistics channel construction $n$ times, the evolutionary game payment matrix is shown in Table 2.

**(2) Stabilization strategy.**   According to the stability principle of differential equations, the conditions of replicating the stable point of dynamic equation is $F(\alpha) = 0$, $f(x) = dF(x)/dx < 0$, the stability of participants in logistics channel construction and their evolution paths are analyzed.

Suppose that in the process of the evolutionary game, regional internal logistics enterprises ($I$) get expected benefits $U'_{I1}$ from implementing 'active cooperation' strategy, $U'_{I2}$ with implementation of the 'passive cooperation' strategy, average expected income $\bar{U}'_I$, according to the

**Table 2. Game payment matrix of logistics channel enterprises guided by government.**

| | | Regional external logistics enterprises | |
|---|---|---|---|
| | | Active cooperation $\beta$ | Passive cooperation $(1-\beta)$ |
| Regional internal logistics enterprises | Active cooperation $\alpha$ | $Q_1 + p[(1+\delta)^{n-1} S - (1-\varepsilon)(C_1 + C_2)];$ $Q_2 + (1-p)[(1+\delta)^{n-1} S - (1-\varepsilon)(C_1 + C_2)]$ | $Q_1 + \lambda S - p(1-\varepsilon)[C_1 + C_2];$ $Q_2 + R_2 - \lambda S - Z$ |
| | Passive cooperation $(1-\alpha)$ | $Q_1 + R_1 - \lambda S - Z;$ $Q_2 + \lambda S - (1-p)(1-\varepsilon)[C_1 + C_2]$ | $Q_1;$ $Q_2$ |

income matrix, we get:

$$U'_{I1} = \beta[Q_1 + p((1+\delta)^{n-1} S - (1-\varepsilon)(C_1 + C_2))] + (1-\beta)[Q_1 + \lambda S - p(1-\varepsilon)[C_1 + C_2]] \quad (9)$$

$$U'_{I2} = \beta[Q_1 + R_1 - \lambda S - Z] + (1-\beta)Q_1 \quad (10)$$

$$\bar{U}'_I = \alpha U'_{I1} + (1-\alpha)U'_{I2} \quad (11)$$

Therefore, the dynamic replication differential equation of logistics enterprises inside the region is as follows:

$$F(\alpha) = d\alpha/dt = \alpha(1-\alpha)(\bar{U}'_I - U'_{I1}) =$$
$$\alpha(1-\alpha)[\beta(pS(1+\delta)^{n-1} - R_1 + Z) + \lambda S - p(1-\varepsilon)(C_1 + C_2)] \quad (12)$$

Similarly, logistics enterprises outside the region ($O$) get expected benefits from implementing 'active cooperation' strategy $U'_{O1}$, $U'_{O2}$ with the implementation of the 'passive cooperation', average expected income $\bar{U}'_O$, according to the income matrix, we get:

$$U'_{O1} = \alpha[Q_2 + (1-p)((1+\delta)^{n-1} S - (1-\varepsilon)(C_1 + C_2))] +$$
$$(1-\alpha)[Q_2 + \lambda S - (1-p)(1-\varepsilon)(C_1 + C_2)] \quad (13)$$

$$U'_{O2} = \alpha[Q_2 + R_2 - \lambda S - Z] + (1-\alpha)Q_2 \quad (14)$$

$$\bar{U}'_O = \beta U'_{O1} + (1-\beta)U'_{O2} \quad (15)$$

Therefore, the replication dynamic differential equation of logistics enterprises outside the region is as follows:

$$F(\beta) = d\beta(t)/dt = \beta(1-\beta)(\bar{U}'_O - U'_{O1}) =$$
$$\beta(1-\beta)[\alpha((1-p)S(1+\delta)^{n-1} - R_2 + Z) + \lambda S - (1-p)(1-\varepsilon)(C_1 + C_2)] \quad (16)$$

According to the Lyapunov stability theory, the Jacobian matrix is used to determine the government participation in the evolution system to achieve the Pareto optimal equilibrium condition $O_4(1, 1)$. The Jacobian matrix $J'$ of logistics channel construction under government

guidance is as follows:

$$
J' = \begin{bmatrix}
(1-2\alpha)[\beta(pS(1+\delta)^{n-1} - R_1 + Z) + & \\
\lambda S - p(1-\varepsilon)(C_1 + C_2)] & \alpha(1-\alpha)(pS(1+\delta)^{n-1} - R_1 + Z) \\
& (1-2\beta)[\alpha((1-p)S(1+\delta)^{n-1} - R_2 + Z) + \\
\beta(1-\beta)((1-p)S(1+\delta)^{n-1} - R_2 + Z) & \lambda S - (1-p)(1-\varepsilon)(C_1 + C_2)]
\end{bmatrix}
$$

Substituting $O_4(1, 1)$ into the Jacobian matrix of government guidance system to obtain the eigenvalue matrix $J_1'$, we get:

$$
J_1' = \begin{bmatrix}
-(pS(1+\delta)^{n-1} - R_1 + Z) - & \\
\lambda S + p(1-\varepsilon)(C_1 + C_2)] & 0 \\
& -((1-p)S(1+\delta)^{n-1} - R_2 + Z) - \\
0 & \lambda S + (1-p)(1-\varepsilon)(C_1 + C_2)
\end{bmatrix}
$$

Because all eigenvalues of the Jacobian matrix are negative, its equilibrium point is an evolutionary stable point, which satisfies the conditions $R_1 - \lambda S - Z < pS(1+\delta)^{n-1} - p(1-\varepsilon)(C_1 + C_2)$ and $R_2 - \lambda S - Z < (1-p)S(1+\delta)^{n-1} - (1-p)(1-\varepsilon)(C_1 + C_2)$, $O_4(1, 1)$ becomes the evolutionary stable point of logistics channel construction under the guidance of the government. According to the above conditions, it is clear that the factors influencing the participation strategy of logistics channel construction under the guidance of the government not only include the market mechanism factors, but also have positive relationship with waiver of the cooperative penalty amount $Z$ and a negative relationship with the active cooperation reward ratio $\varepsilon$. $R_1 - \lambda S - Z < pS(1+\delta)^{n-1} - p(1-\varepsilon)(C_1 + C_2)$ indicates that the active cooperation of logistics enterprises inside the region under the guidance of the government can secure higher profits. $R_1 - \lambda S - Z$ denotes the net income value that the regional logistics enterprises give up under cooperation with the government incentive mechanism, and $pS(1+\delta)^{n-1} - p(1-\varepsilon)(C_1 + C_2)$ denotes the net income value of active cooperation among logistics enterprises inside the region under the government incentive mechanism. Correspondingly, under the guidance of the government, enterprises along the logistics channel actively cooperate with each other when net income $(1-p)S(1+\delta)^{n-1} - (1-p)(1-\varepsilon)(C_1 + C_2)$ is greater than the net income of abandonment cooperation $R_2 - \lambda S - Z$. Under these conditions, the participants will give priority to the active cooperation strategy. Therefore, the government reward and punishment mechanism can not only reduce the income of the party that gives up active cooperation, but also can also reduce the cooperation cost. Compared with the market model, the participation cost of logistics channel construction, guided by the government is lower, especially for the small and medium-sized logistics enterprises along the channel, which have a high degree of dependence on government funds. The low-cost advantage after reward becomes the participation incentive for logistics channel construction. In addition, the government reward strategy can improve the positive incentive effect level of active cooperation $\delta$. The high incentive level makes the net income of active cooperation greater than that of giving up cooperation, $R_1 - \lambda S - Z < pS(1+\delta)^{n-1} - p(1-\varepsilon)(C_1 + C_2)$ and $R_2 - \lambda S - Z < (1-p)S(1+\delta)^{n-1} - (1-p)(1-\varepsilon)(C_1 + C_2)$ can be satisfied more easily. However, the government's punishment makes the participants give up the active strategy to get a lower net income, so the logistics channel participants tend to choose the active cooperation strategy.

## 4. Numerical experiments and simulation results analysis

Aiming at the construction mode of a national unified large market logistics channel under market mechanisms and government guidance, the evolution path of the system game is simulated by software MatlabR2018a. To verify the feasibility and effectiveness of the model, on the basis of satisfying the parameter constraint relationship of part 2.1 and the practice of logistics channel construction, this paper sets the basic parameters and the initial test value and specific range of the model in a random way, which can avoid the influence of specific examples on the credibility of the model performance. In this part, firstly, through numerical simulation, the influence of reward parameters and penalty amount under the guidance of the government on the game strategy of participating enterprises is analyzed, the sensitivity analysis is used to clarify the influence mechanism of the initial participation intention, positive cooperation benefit, cost proportion coefficient, penalty coefficient and construction cost in the two modes.

### 4.1 Research on the effectiveness of reward and punishment mechanism under the guidance of the government

**(1) The effectiveness analysis of government reward parameters on both sides' game strategies.**    According to the principle of assignment parameters, we set $\alpha = 0.5$, $\beta = 0.5$, $p = 0$, $\lambda = 0.45$, $\delta = 0.1$, $S = 11$, $C_1 = 6$, $C_2 = 3$, $Q_1 = 10$, $Q_2 = 10$, $R_1 = 9$, $R_2 = 9$, $Z = 0$. The reward proportion of active cooperation under the guidance of the government $\varepsilon$ has assignment range $0.1 \leq \varepsilon \leq 0.9$, and the step size is 0.1. As can be seen from the game evolution strategy, the influence effect of logistics channel construction participants in Figs 1 and 2, when $\varepsilon \leq 0.3$, logistics enterprises along the logistics channel all choose the strategy of 'passive cooperation'; when $\varepsilon = 0.4$ and $\varepsilon = 0.5$, the logistics enterprises inside the region choose the'passive cooperation' strategy and the enterprises outside the region choose the'active cooperation' strategy; when $\varepsilon \geq 0.6$, enterprises along the logistics channel all choose the strategy of 'active cooperation'. The step size of the government-guided reward ratio $\varepsilon$ in Figs 1 and 2 is the same, according to the evolutionary game effect, the marginal effect of the increasing government

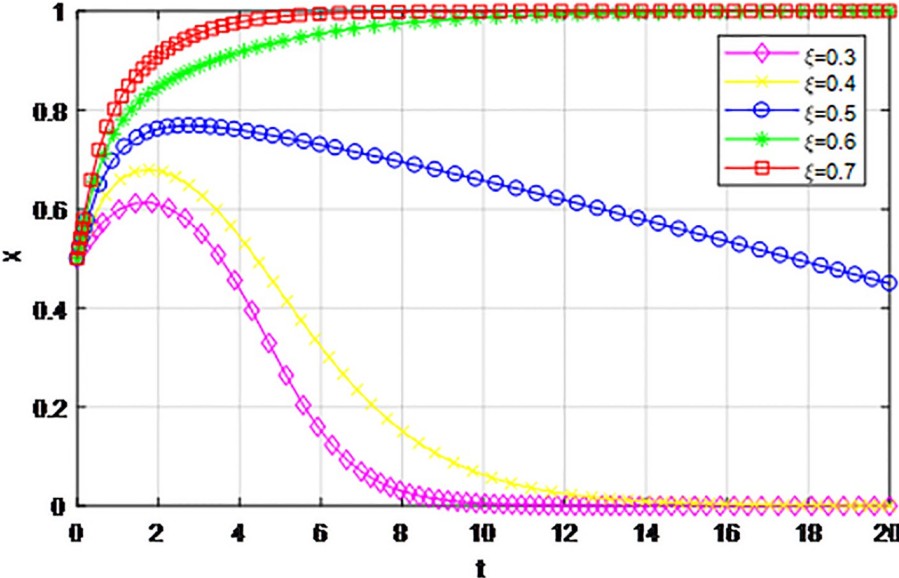

**Fig 1. The impact of proportion of government awards $\varepsilon$ on logistics enterprises in the region.**

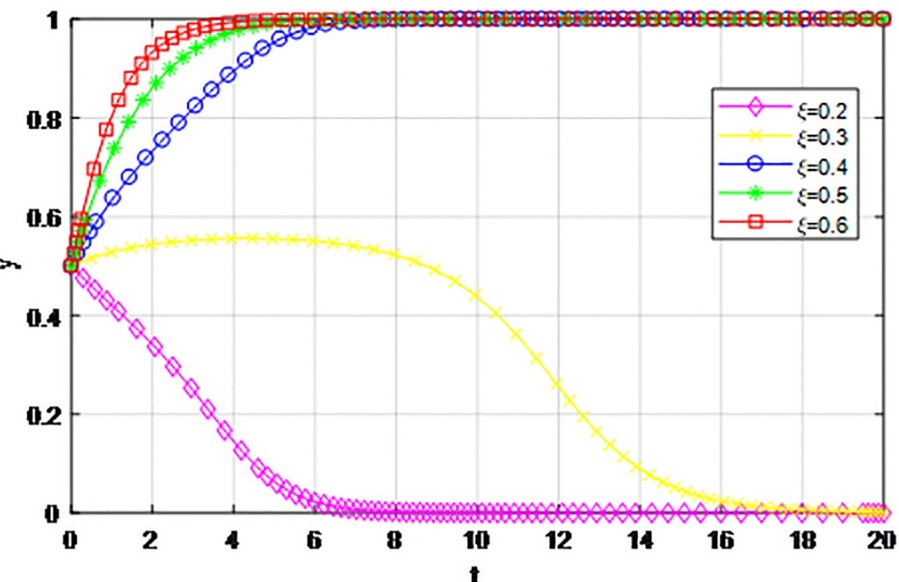

**Fig 2. The impact of proportion of government awards $\varepsilon$ on logistics enterprises outside the region.**

reward ratio $\varepsilon$ gradually decreases, indicating that the government can promote the implementation of the reward system for logistics channel participants under the unified national market, but when $\varepsilon$ increases, its positive incentive effect continuously decreases. Therefore, according to the practice of building the logistics channel in the national unified market, the government can give some positive incentives to the participants through policy inclination, but from the perspective of the optimal adjustment effect, the reward amount does not need to entail excessive financial cost.

The following graph (named Figs 1 and 2) can also be produced:

According to the principle of assignment parameters, we set $\alpha = 0.5$, $\beta = 0.5$, $p = 0.45$, $\lambda = 0.45$, $\delta = 0.1$, $S = 11$, $C_1 = 6$, $C_2 = 3$, $Q_1 = 10$, $Q_2 = 10$, $R_1 = 9$, $R_2 = 9$, $\varepsilon = 0$. Under the guidance of the government, the penalty amount for the party giving up cooperation is $Z$. Setting penalty amount $Z$ as 1, 2, 3, 4, and 5, respectively, the influence effect of the evolution game strategy of the logistics channel system is shown in Figs 3 and 4. When $Z = 1$, enterprises along the logistics channel all choose the strategy of 'passive cooperation'; when $Z = 2$, the marginal effect of logistics channel participants is significant, the logistics enterprises in the region choose the strategy of 'passive cooperation', and the enterprises outside the region finally choose the strategy of 'active cooperation' after the delay evolution. When $Z \geq 4$, the enterprises along the logistics channel all finally choose the strategy of 'active cooperation', and with the increase of the government's penalty amount, the negative reinforcement effect is more obvious, and the two sides realize the active cooperation equilibrium faster. Therefore, in the practice of logistics channel construction, the government's punishment measures for abandoning cooperation can not only prevent moral hazard for both sides, but also help participants to achieve positive cooperation and balanced efficiency, and ensure the steady progress of logistics channel evolution.

The following graph (named Figs 3 and 4) can also be produced:

**(2) Sensitivity analysis of important parameters of logistics channel under two modes.** *1) Impact of initial willingness to participate on the game strategy.* Based on the effectiveness analysis of the reward and punishment mechanism under the guidance of the

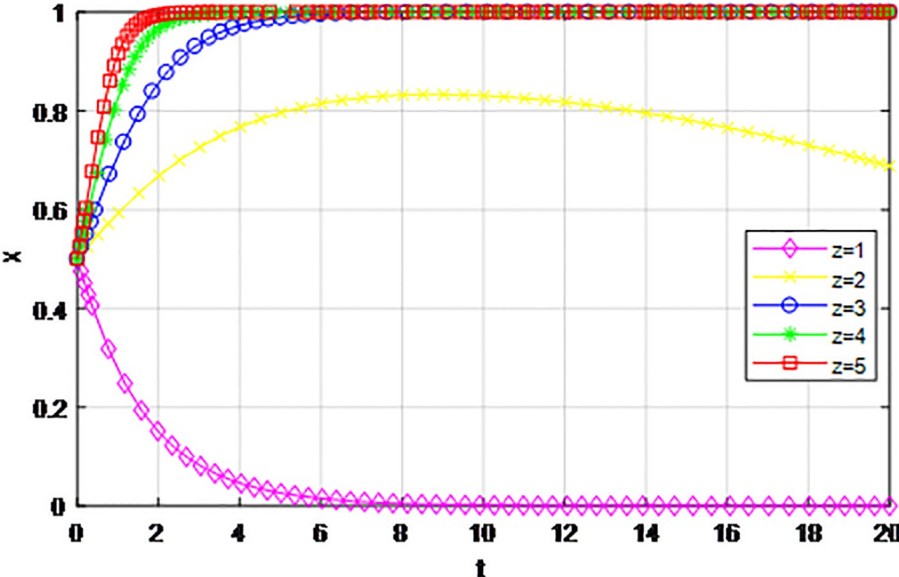

**Fig 3. Impact of government punishment Z on logistics enterprises inside the region.**

government, the government rewards the proportion $\varepsilon = 0.6$, the penalty amount $Z = 4$, and the other parameters are $p = 0.45$, $\lambda = 0.45$, $\delta = 0.1$, $S = 11$, $C_1 = 6$, $C_2 = 3$, $Q_1 = 10$, $Q_2 = 10$, $R_1 = 9$, $R_2 = 9$, respectively. The evolution trend of the market mechanism and the government-guided logistics channel construction participants under changes to initial participation willingness are analyzed. First of all, under the guidance of the market mechanism and government, when the initial willingness of logistics enterprises outside the region to participate in channel construction is low, the order $\beta = 0.2$. If the initial participation intention of logistics enterprises inside the region ($\alpha = 0.2, 0.4, 0.6, 0.8$) changes, the dynamic evolution process of

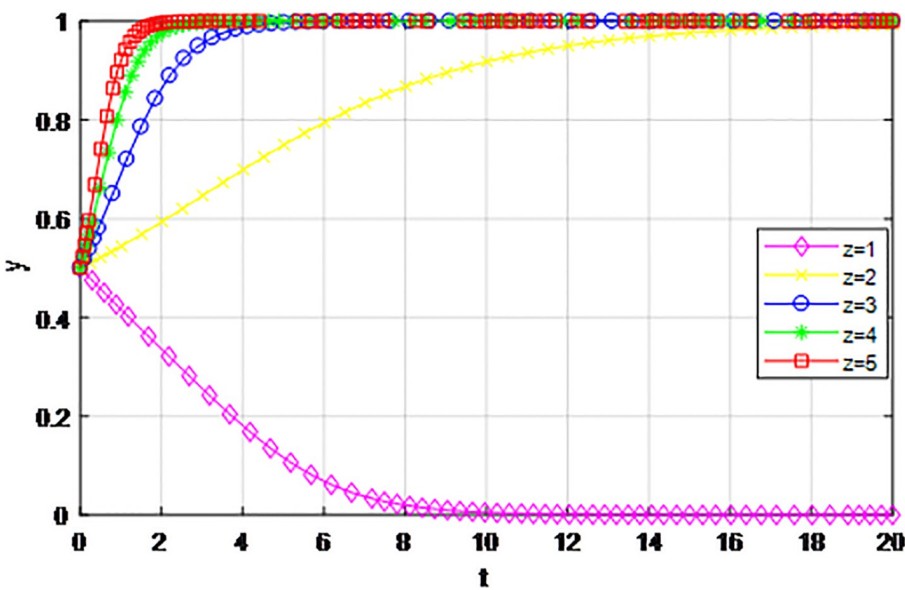

**Fig 4. Impact of government punishment Z on logistics enterprises outside the region.**

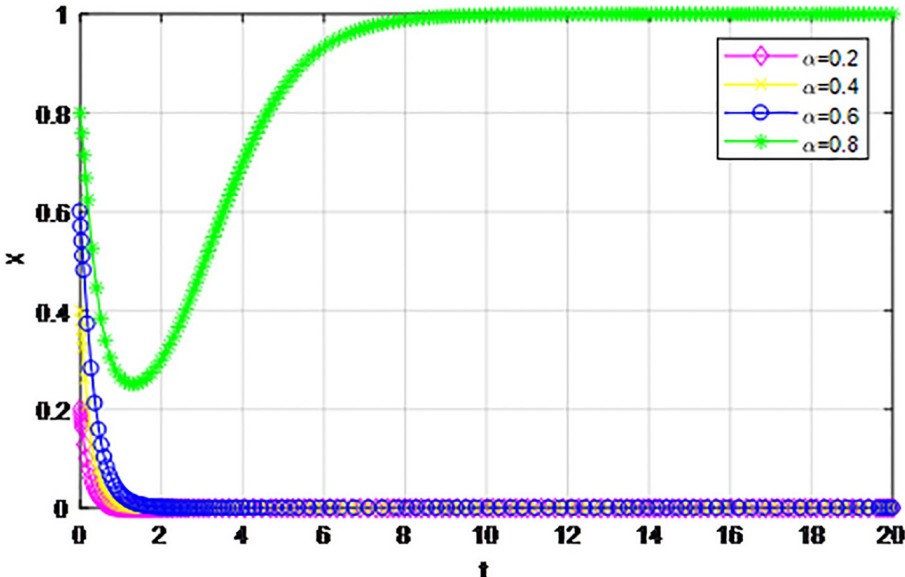

**Fig 5. Evolution of logistics enterprise strategy in the region under the market mechanism $\beta = 0.2$.**

market mechanism and government-guided logistics channel participation strategy is shown in Figs 5 and 6. As can be seen from the two figures, under the market mechanism, only when the initial participation willingness of logistics enterprises in the region is high (i.e.$\alpha = 0.8$), does the probability of choosing an 'active cooperation' strategy converge to 1, and when the initial intention of logistics enterprises inside the region is low (i.e.$\alpha = 0.2, 0.4, 0.6$), the probability of choosing the 'active cooperation' strategy finally converges to 0. The higher the willingness to give up, the faster the evolution speed of the system. It can be seen that government

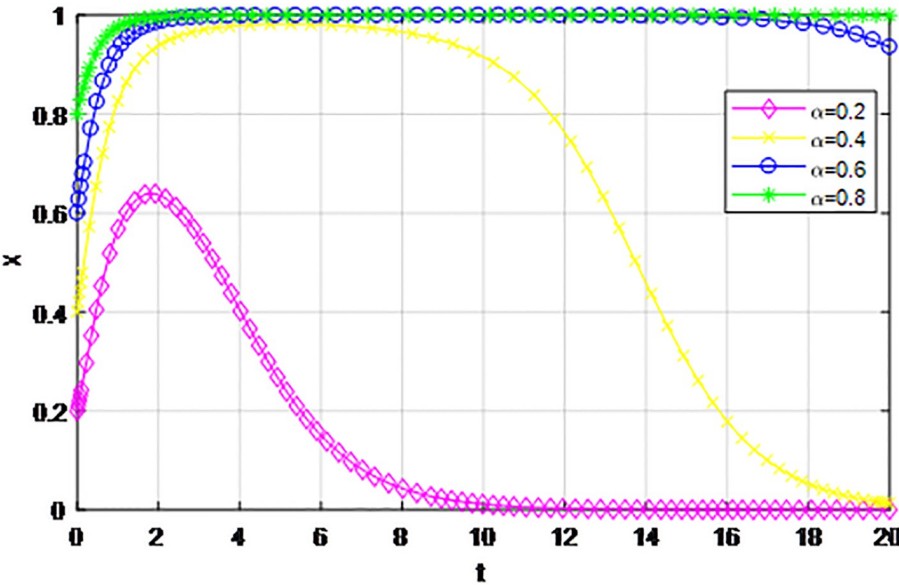

**Fig 6. Evolution of logistics enterprise strategy in the region under the guidance of the government $\beta = 0.2$.**

guidance has significantly improved the evolution trend of logistics channels, and reduced the convergence rate of logistics enterprises inside the region when their initial intentions are low. Secondly, we compare and analyze the game trend of system evolution when logistics enterprises outside the region have high initial willingness to participate, and make $\beta = 0.8$. Similarly, the initial participation willingness of logistics enterprises inside the region ($\alpha = 0.2, 0.4, 0.6, 0.8$) changes, the dynamic evolution process of market mechanisms and government-guided logistics channel participation strategy is shown in Figs 7 and 8. According to the two figures, it is shown that the market mechanism and the probability of logistics enterprises inside the region choosing an 'active cooperation' strategy under the guidance of the government eventually converge to 1. Under the market model, the higher the participation willingness of logistics enterprises in the region, the faster the convergence speed of the logistics channel construction evolution system. Compared with the market mechanism, the system evolution efficiency under the guidance of the government is higher, and the logistics enterprises converge to the 'active cooperation' strategy in a short time, and it has high stability. The policy implication is that the government should strengthen the role of reward and punishment mechanisms, and the cooperation balance of the evolutionary system can be realized by increasing the initial participation willingness of logistics enterprises inside the region.

The following graph (named Figs 5–8) can also be produced:

*2) Impact of changing earnings from active cooperation S on game strategy.* This section discusses the influence of changing benefits of active cooperation $S$ on the evolution trend of participants in the market mechanism and government-guided logistics channel construction. We set $\alpha = 0.5$, $\beta = 0.5$, $p = 0.45$, $\lambda = 0.45$, $\delta = 0.1$, $C_1 = 6$, $C_2 = 3$, $Q_1 = 10$, $Q_2 = 10$, $R_1 = 9$, $R_2 = 9$, $\varepsilon = 0$, $Z = 4$, and suppose $S = 8, 10, 12, 14$. The evolution process of logistics channel participants' strategy under the market mechanism is shown in Figs 9 and 10, and the evolution process of logistics channel participants' strategy under government guidance is shown in Figs 11 and 12. As shown in Figs 9 and 10, under the market mechanism, when the positive cooperative income $S = 8$, the logistics channel participants choose the strategy of 'passive cooperation'. When $S = 10, 12$, the logistics enterprises inside the region finally choose the strategy of

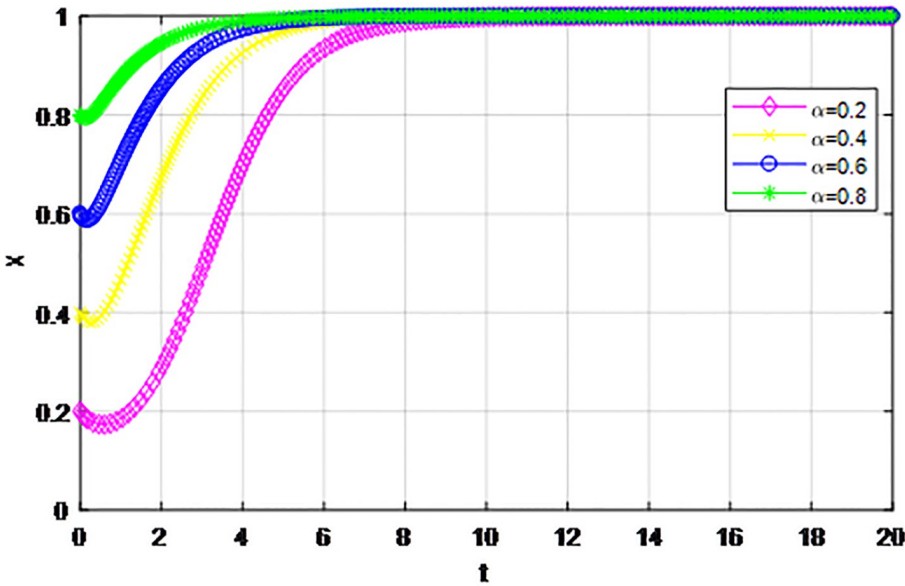

**Fig 7. Evolution of logistics enterprise strategy in the region under the market mechanism $\beta = 0.8$.**

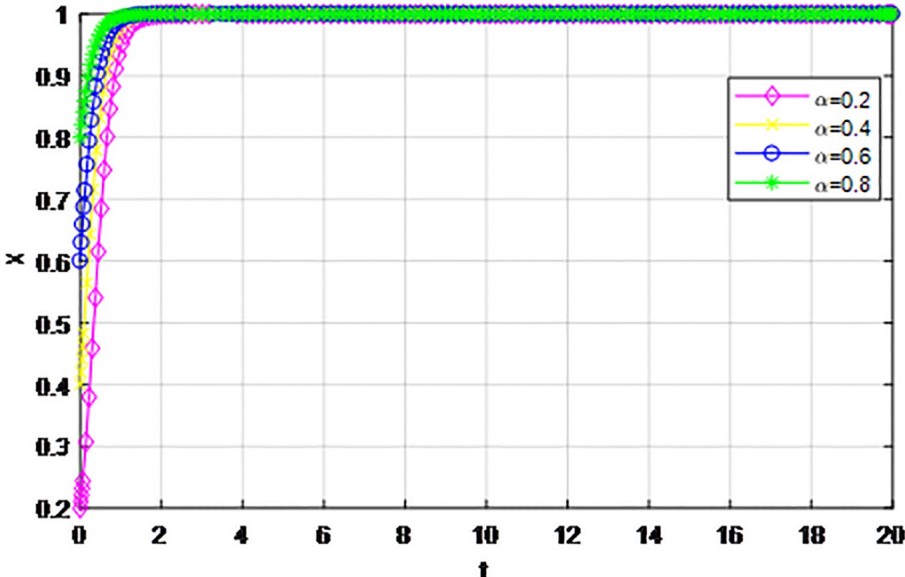

**Fig 8. Evolution of logistics enterprise strategy in the region under the guidance of the government $\beta$ = 0.8.**

'active cooperation', while the enterprises outside the region choose the strategy of 'passive cooperation'. Finally, the benefits of active cooperation are further improved to $S$ = 14, then both sides choose the strategy of 'active cooperation'. Therefore, with other conditions remaining unchanged, increasing the active cooperation gains $S$ has a significant effect of promoting the system-evolution equilibrium, and improving the benefits of active cooperation will prompt the participants to shift their passive cooperative attitude to active participation.

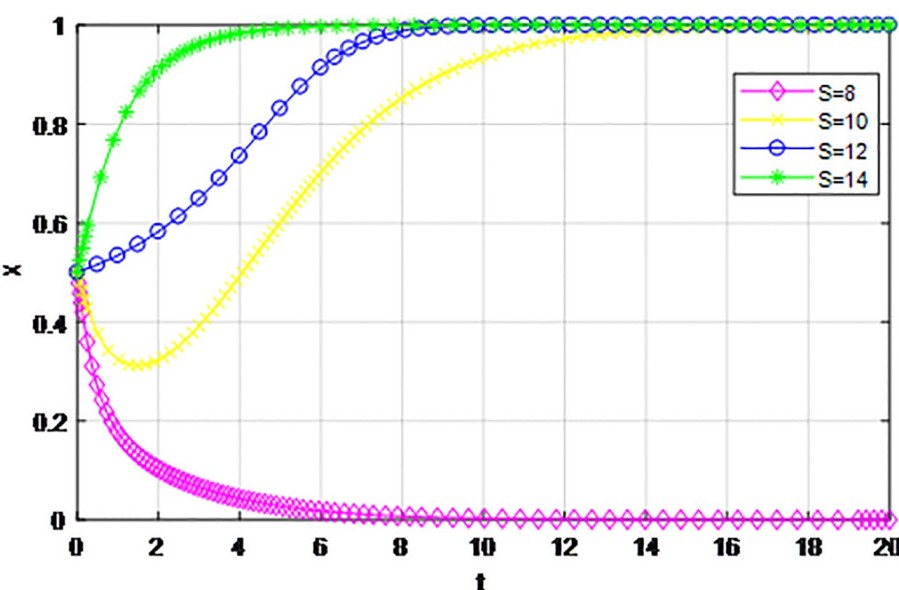

**Fig 9. The evolution of enterprise strategy inside the region when the cooperative income $S$ changes under the market mechanism.**

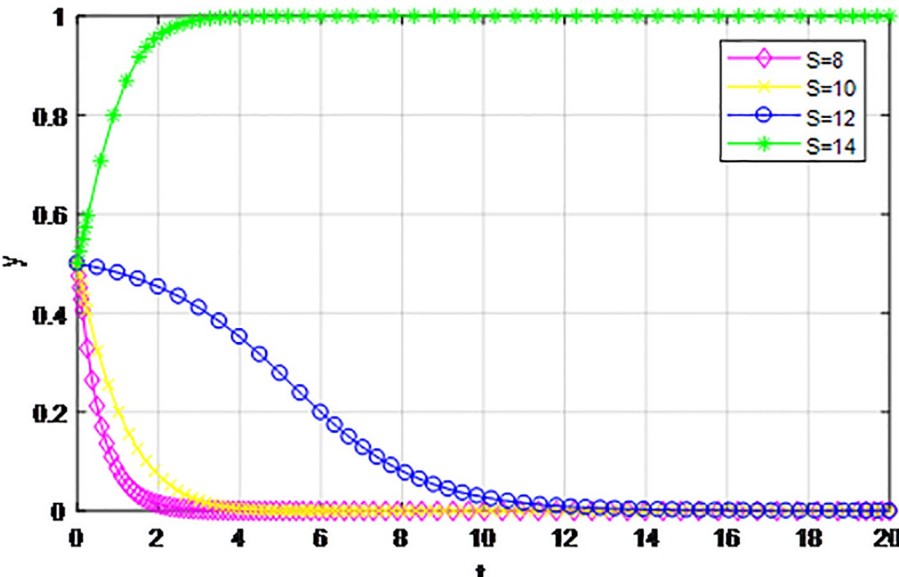

**Fig 10. The evolution of enterprise strategy outside the region when the cooperative income *S* changes under the market mechanism.**

Therefore, if the regional logistics channel construction project under the national unified large market strategy has a good prospect and a considerable expected income in the future, it can reach the cooperative equilibrium state of the participants in the evolutionary game system. Compared with the market mechanism, government guidance can lower the threshold of evolutionary equilibrium. When active cooperation benefits *S* = 8, the logistics channel of both

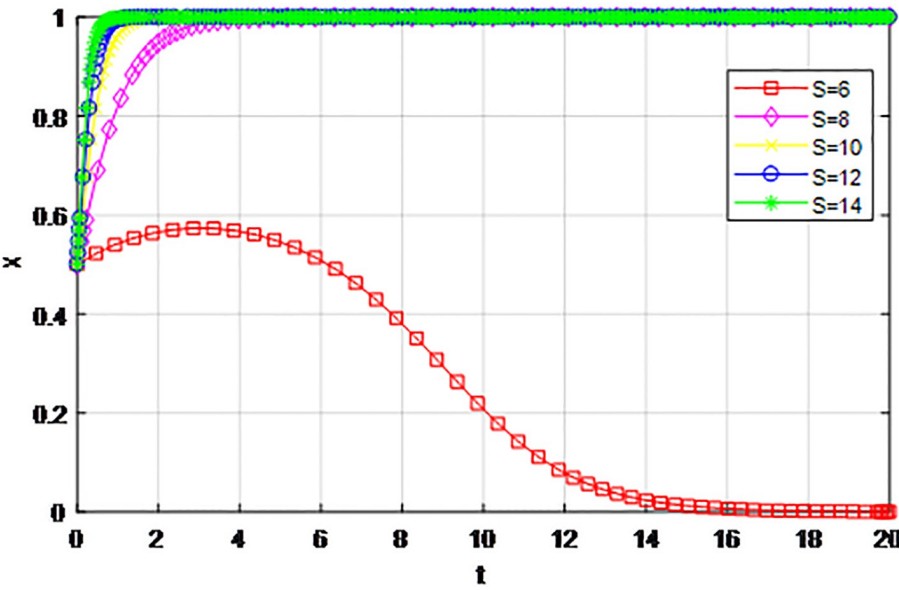

**Fig 11. The evolution of enterprise strategy inside the region when the cooperation income *S* changes under the guidance of the government.**

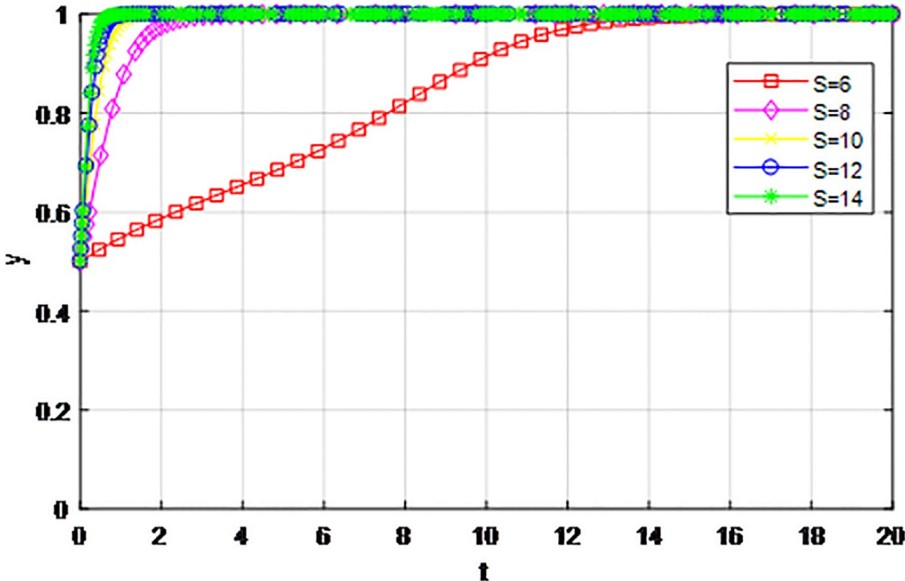

**Fig 12. The evolution of enterprise strategy outside the region when the cooperation income *S* changes under the guidance of the government.**

sides choose the strategy of 'active cooperation', that is, the two sides can reach cooperation when the benefits of active cooperation are low. At the same time, government guidance also improves the evolution efficiency of the system, and the participants can make the final equilibrium decision in a short time.

The following graph (named Figs 9–12) can also be produced:

*3) Impact of cost proportion coefficient p on strategy evolution.* In this section, we discuss the impact of changes in the cost proportion coefficient *p* on the evolution trend of participants in the market mechanism and government-guided logistics channel construction. We set $\alpha = 0.5$, $\beta = 0.5$, $S = 11$, $\lambda = 0.45$, $\delta = 0.1$, $C_1 = 6$, $C_2 = 3$, $Q_1 = 10$, $Q_2 = 10$, $R_1 = 9$, $R_2 = 9$, $\varepsilon = 0$, $Z = 4$. Suppose $p = 0.2, 0.4, 0.6, 0.8$, the evolution process of logistics channel construction participants' strategies under market the mechanism is shown in Figs 13 and 14, and the evolution process of participants' strategies under government guidance is shown in Figs 15 and 16. As shown in Figs 13 and 14, with the current cost ratio coefficient under the market mechanism $p = 0.2, 0.4$, the enterprises inside the region participating the logistics channel finally choose the 'active cooperation' strategy, while the enterprises outside the region correspondingly choose the 'passive cooperation' strategy; Under the market mechanism, when the proportion coefficient of current cost $p = 0.6, 0.8$, the logistics channel participants make opposite strategic decisions. Therefore, when the participants pay lower costs, the corresponding logistics enterprises will implement an active cooperation strategy, which is in line with the conclusion of fairness of cost-sharing. The proportion coefficient of logistics channel construction cost under government guidance *p* has no obvious influence on the final strategy choice of the evolutionary system. All the participating enterprises finally choose the 'active cooperation' strategy, and the game has no substantial change in the end. Therefore, against the background of a unified national market, when designing the logistics channel construction scheme, it is necessary to incorporate the influence effect of the cost proportion coefficient on the evolution game of the participants, scientifically formulate the top-level system, ensure reasonable cost

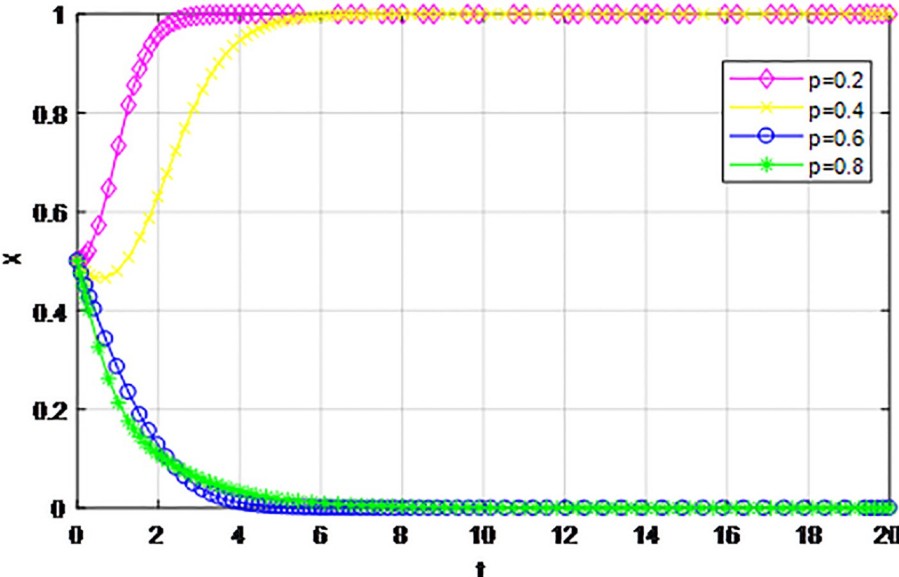

**Fig 13. Enterprise strategy evolution inside the region when the cost ratio $p$ changes under the market mechanism.**

sharing, mobilize the enthusiasm of both parties, and realize the fair distribution of the system cost.

The following graph (named Figs 13–16) can also be produced:

*4) Impact of penalty coefficient $\lambda$ on strategy evolution.* This section discusses the impact of changes in the penalty-coefficient $\lambda$ on the evolution trend of participants in the market

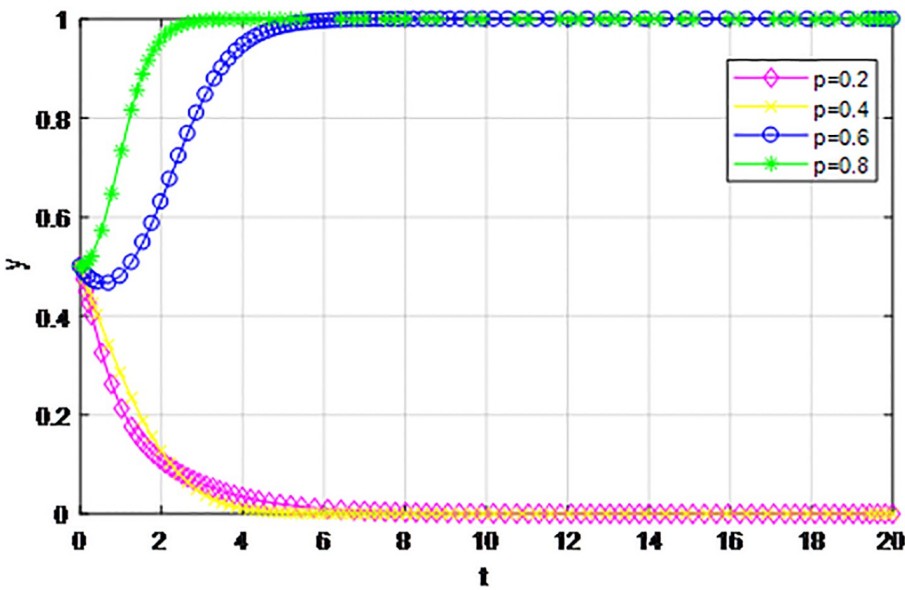

**Fig 14. Enterprise strategy evolution outside the region when the cost ratio $p$ changes under the market mechanism.**

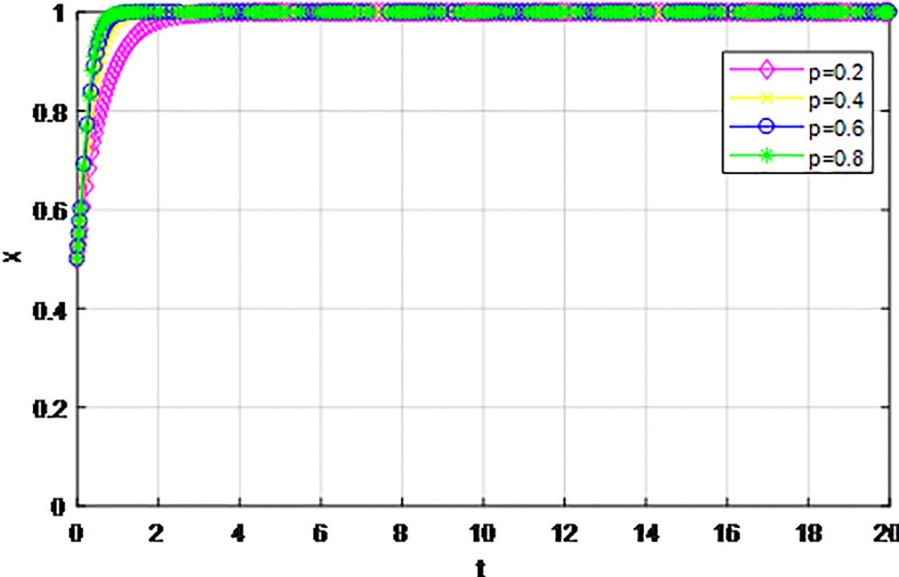

**Fig 15. Enterprise strategy evolution inside the region when the cost ratio *p* changes under the guidance of the government.**

mechanism and government-guided logistics channel construction. We set $\alpha = 0.5$, $\beta = 0.5$, $p = 0.45$, $S = 11$, $\delta = 0.1$, $C_1 = 6$, $C_2 = 3$, $Q_1 = 10$, $Q_2 = 10$, $R_1 = 9$, $R_2 = 9$, $\varepsilon = 0.6$, $Z = 4$. Suppose $\lambda = 0.1, 0.3, 0.5, 0.7, 0.9$, Figs 17 and 18 show the evolution process of logistics channel construction participants' strategies under the market mechanism, and Figs 19 and 20 show the evolution process of logistics channel construction participants' strategies under government

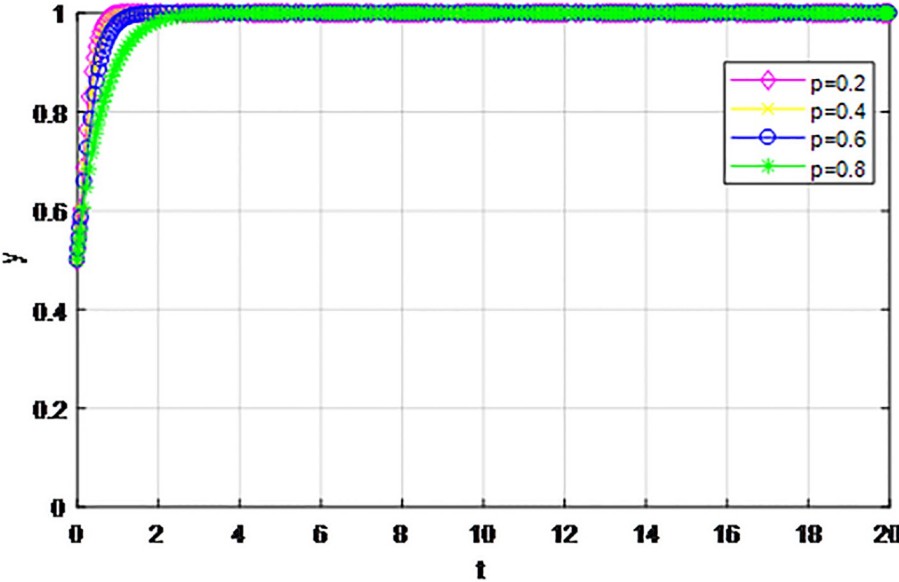

**Fig 16. Enterprise strategy evolution outside the region when the cost ratio *p* changes under the guidance of the government.**

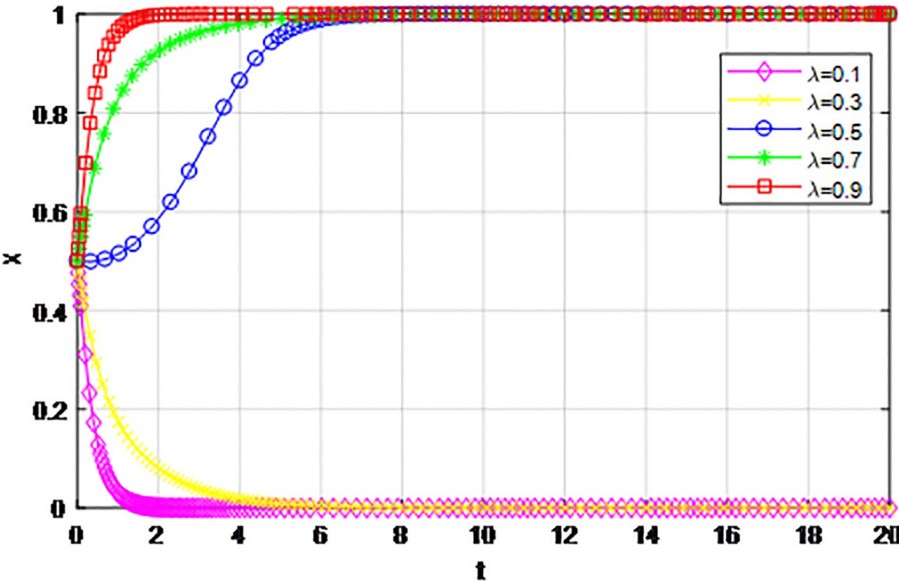

**Fig 17. The evolution of enterprise strategy inside the region when the penalty coefficient $\lambda$ changes under market mechanism.**

guidance. As shown in Figs 17 and 18, when the penalty coefficient under the market mechanism $\lambda$ = 0.1, 0.3, the logistics channel participants eventually choose the 'passive cooperation' strategy. When the penalty coefficient $\lambda$ = 0.5, the enterprises in the logistics channel region choose the strategy of 'active cooperation', the enterprises outside the region choose the strategy of 'passive cooperation'. When the penalty coefficient $\lambda$ = 0.7, 0.9, both sides of the logistics

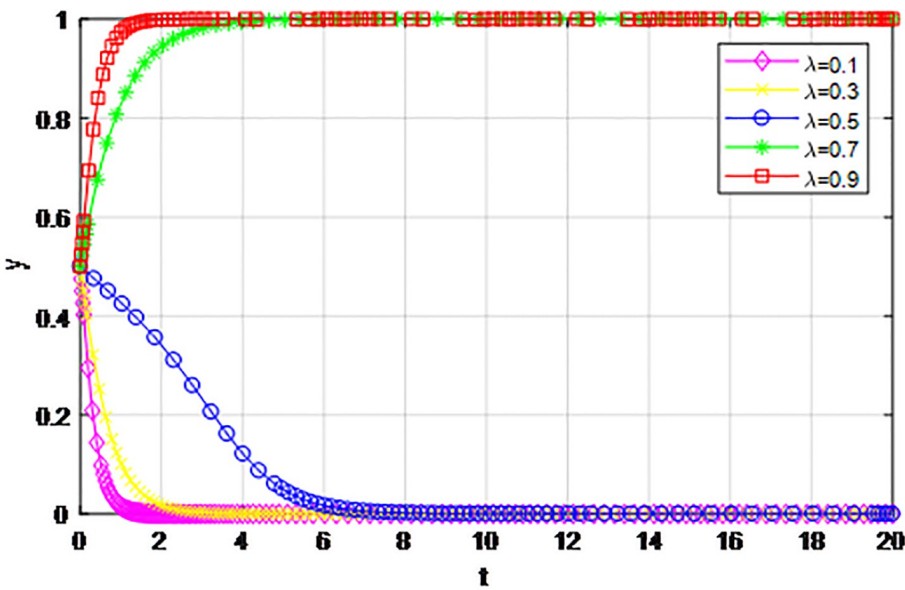

**Fig 18. The evolution of enterprise strategy outside the region when the penalty coefficient $\lambda$ changes under market mechanism.**

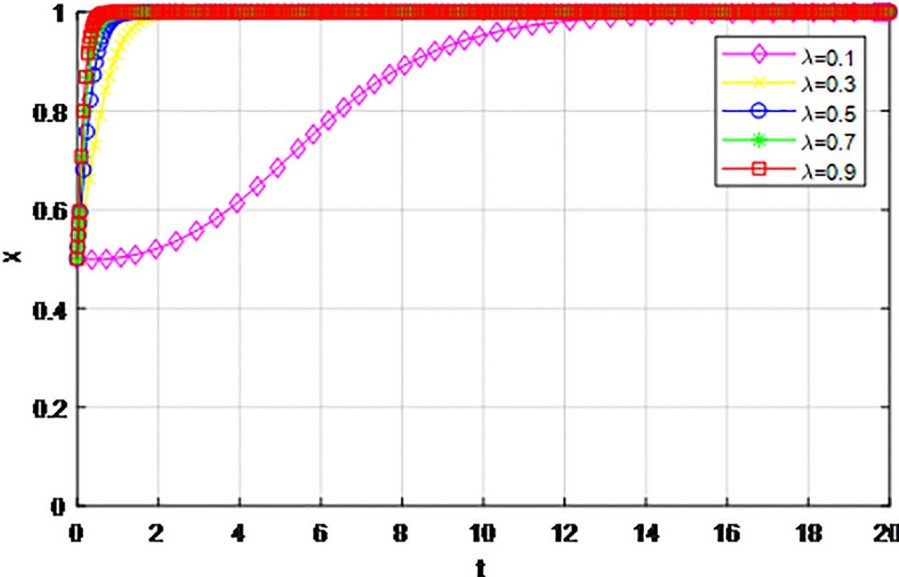

**Fig 19. The evolution of enterprise strategy inside the region when the penalty coefficient $\lambda$ changes under the guidance of government.**

channel choose the strategy of 'active cooperation', and the higher the penalty coefficient, the faster the evolution rate. Thus, the penalty coefficient $\lambda$ under the market model has a significant impact on both parties' evolutionary strategies, and increasing the penalty coefficient can effectively promote both parties' participation in positive decision-making. Under the guidance of the government, only when the penalty coefficient $\lambda = 0.1$, will both sides of the logistics channel choose the strategy of 'active cooperation', and the evolution rate of the

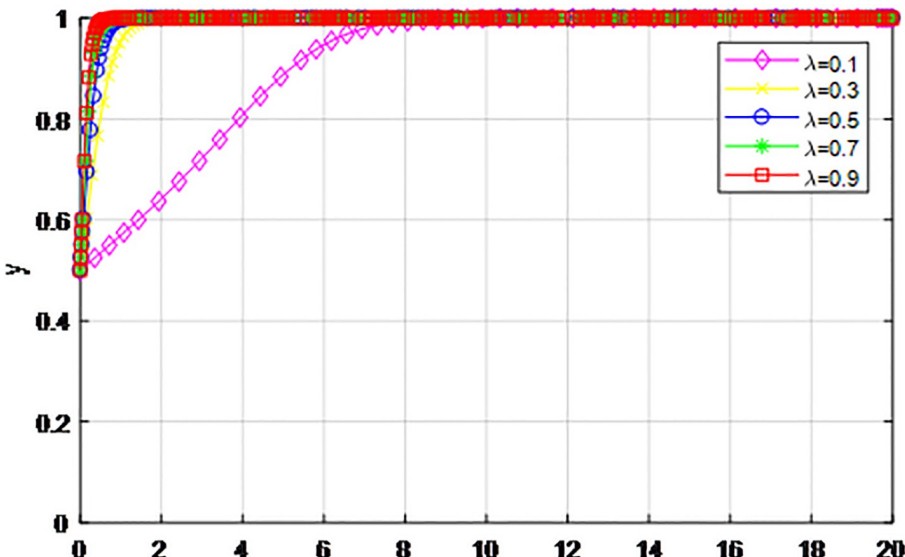

**Fig 20. The evolution of enterprise strategy outside the region when the penalty coefficient $\lambda$ changes under the guidance of government.**

improvement coefficient is accelerating, that is, under the government mode, the penalty coefficient obviously improves the strategic choice of both sides, and a lower penalty coefficient can achieve evolutionary equilibrium. Therefore, in practice, the penalty coefficients of different modes of logistics channels need to be designed to fall within a reasonable range. Under the market mode, a lower penalty coefficient increases the moral hazard of the participants, which eventually leads to the failure of cooperation. Under the guidance of the government, a higher coefficient can improve the evolution rate, but in the long run, the mandatory government system has a negative impact on the stability of the multi-stage game system.

The following graph (named Figs 17–20) can also be produced:

*5) Impact of construction cost $C_1$ on game strategy.* This section discusses the influence of changes to the construction cost of logistics channel $C_1$ on the evolution of participants under the market mechanism and government guidance modes. We set $\alpha = 0.5$, $\beta = 0.5$, $s = 0.45$, $\lambda = 0.45$, $\delta = 0.1$, $s = 11$, $C_2 = 3$, $Q_1 = 10$, $Q_2 = 10$, $R_1 = 9$, $R_2 = 9$, $\varepsilon = 0.6$, $Z = 4$. Suppose $C_1 = 6, 7, 8, 9, 10$, the evolution process of participants' strategies in logistics channel construction under market mechanism is shown in Figs 21 and 22, and the evolution process of participants' strategies in logistics channel construction under government guidance is shown in Figs 23 and 24. As shown in Figs 21 and 22, when the construction cost is at a low level under the market mechanism $C_1 = 6$, the logistics channel participants finally choose the strategy of 'active cooperation'; When the construction cost $C_1 = 7$, the enterprises inside the logistics channel region choose the strategy of 'active cooperation', while the enterprises outside the region finally choose the strategy of 'passive cooperation'; When the construction cost increases to $C_1 = 8, 9, 10$, both sides of the logistics channel choose the strategy of 'passive cooperation', and the higher the cost, the faster the abandonment. Under the guidance of the government, the construction cost of logistics channel has a significant effect on the evolution of system strategy, when $C_1 = 6, 7, 8, 9, 10$ the strategic decisions of both parties are 'active cooperation', but when the construction cost continues to increase to 28, the final strategic choice of both parties turns into 'passive cooperation'. That is, when the construction cost exceeds the capacity of the

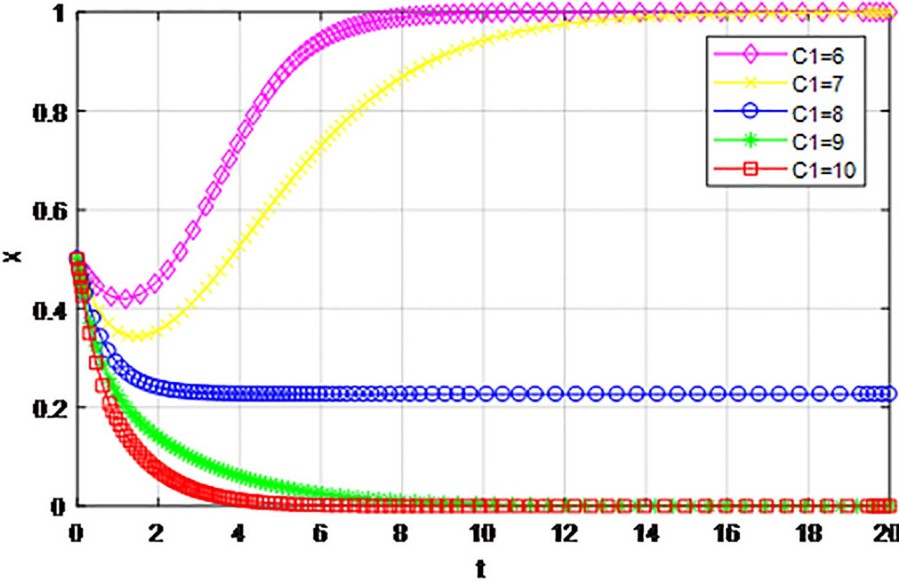

**Fig 21. The evolution of enterprise strategy inside the region when the construction cost $C_1$ changes under market mechanism.**

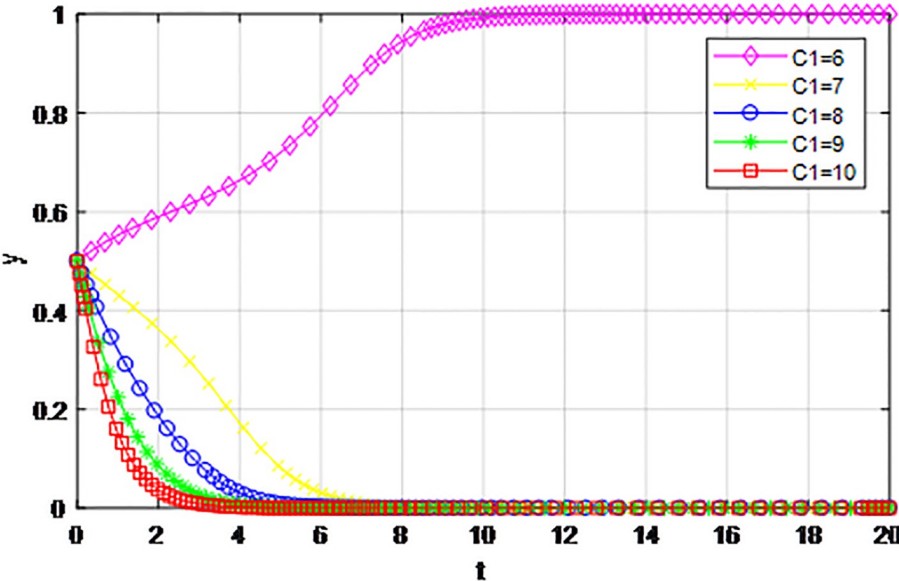

**Fig 22. The evolution of enterprise strategy outside the region when the construction cost $C_1$ changes under market mechanism.**

participants, both parties will inevitably give up cooperation even if the government pays the penalty cost. Therefore, the government should determine the subsidy amount according to the actual situation of regional logistics channels, reduce the payment cost of participants, especially small and medium-sized logistics enterprises, and then realize the balance of cooperation in logistics channel construction.

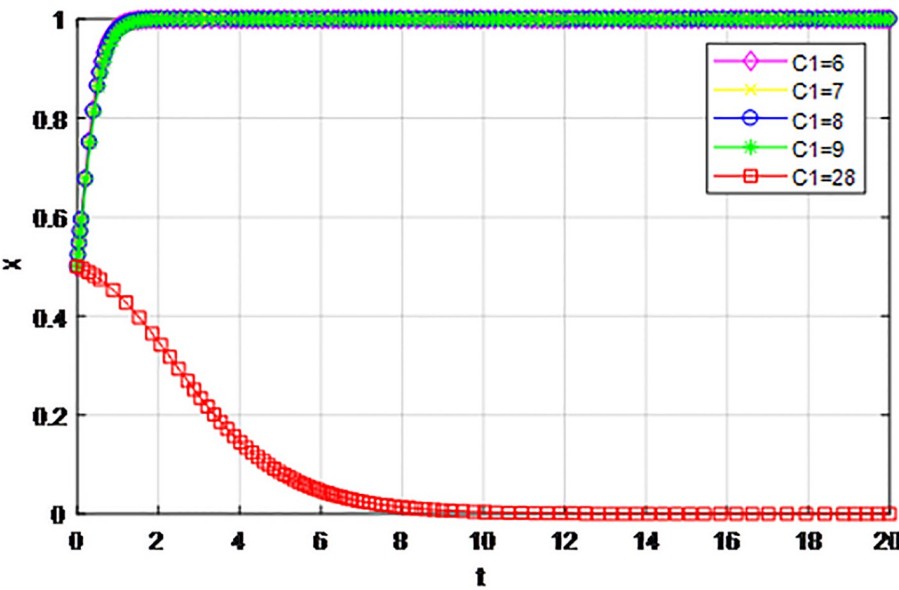

**Fig 23. The evolution of enterprise strategy inside the region when the construction cost $C_1$ changes under the guidance of government.**

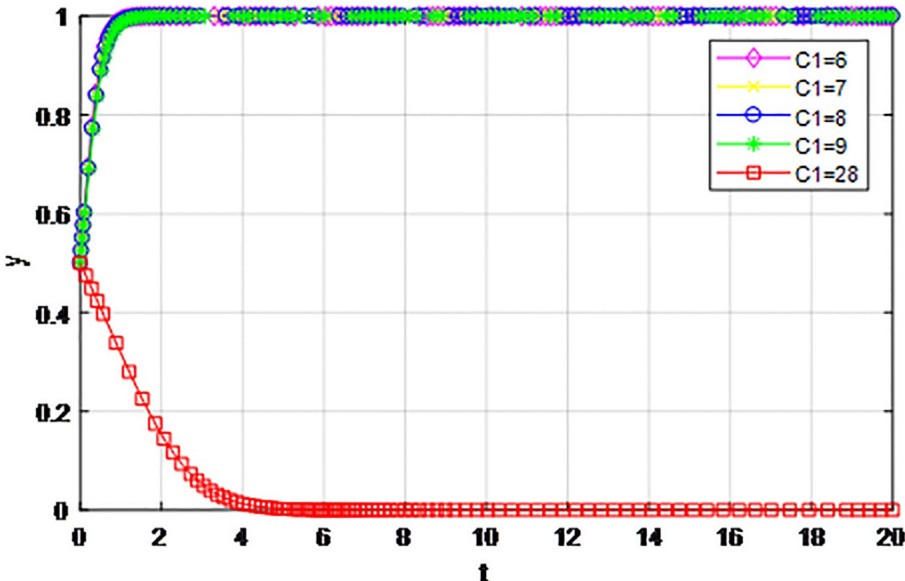

**Fig 24. The evolution of enterprise strategy outside the region when the construction cost $C_1$ changes under the guidance of government.**

The following graph (named Figs 21–24) can also be produced:

## 5. Conclusion and managerial insights

### 5.1 Research conclusion

A convenient and smooth logistics channel is conducive to regional logistics cooperation and resource sharing, and helps the implementation of the national unified market strategy [41, 42]. In order to explore the dynamic process of enterprises' participation in a logistics channel strategy under the guidance of the government, objectively reveal the evolutionary game relationship of participation behaviors of various stakeholders, and provide a theoretical basis for the government to formulate reasonable policies, this paper examines the mechanisms of government guidance on logistics channel construction, and uses an evolutionary game model to explore the dilemma of 'market failure' under the market mechanism. Based on this analysis it examines the conditions for optimizing the cooperative innovation mode under the guidance of the government. Through simulation of logistics channel selection strategies under the two mechanisms, the influence path of key factors on government and enterprise participation behaviors is clarified. The results show that the guidance of the government is more likely to achieve a stable and balanced evolution the game strategy of logistics channel participation behavior, and also that the initial intentions of both parties influence each other. Government participation can change the effects of cooperative income, penalty coefficient and construction cost on the system game strategy, which has a positive effect on logistics channel cooperation [43]. Finally, the government should take reasonable guidance to prevent the negative effects caused by adverse selection.

### 5.2 Managerial insights

The construction of logistics channels is conducive to strengthening the smooth connectivity of the domestic market, promoting the active connection between the domestic market and

the international market, promoting the domestic circulation and circulation of all links of production, distribution, circulation and consumption, and making the smooth flow of commodity elements and resources in a larger scope, which is conducive to the construction of a unified national market. In order to improve the effectiveness of the logistics channel participation strategy of the national unified large market, this study puts forward the following suggestions:

**(1) Formulate scientific incentive policies and take multiple measures to improve the willingness of logistics enterprises to participate.** Through the above evolutionary game and simulation analysis, it is shown that the final stable state of the participants' evolutionary game is influenced by the initial participation willingness of the government and enterprises. Under the market mechanism, the logistics channel construction often faces the problem of 'market failure'. In order to promote the high-quality construction of the national unified market logistics channel, the government has an important role of formulating reward and punishment measures to promote the logistics enterprises along the route to build cross-regional channels. Under the unified national market, we should first strengthen the government's awareness of participation, give full play to the positive incentive effect of guiding measures, effectively update the concept of the new development pattern of logistics enterprises and encourage active participation in logistics channel practice activities by increasing the cooperation income of both parties, strengthening the punishment of the defaulting party and reducing the construction cost. The government has adopted a guiding policy and there have been improvements the initial willingness of enterprises to participate in the construction of logistics channels. By giving full play to the advantages and enthusiasms of various regions, the trans-regional logistics cooperation mechanism, mutual assistance mechanisms and support mechanisms have been gradually improved, and a new pattern of complementary advantages, mutual promotion and common development has been formed.

**(2) The government should create conditions for high-quality cooperation among enterprises and strengthen the process supervision and control mechanism.** Through the above evolutionary game and simulation analysis, we observe that increasing the initial willingness to cooperate and the cooperation income is beneficial in prompting the logistics enterprises of both sides to realize the balance of logistics channel construction. The level of both depends on the quality results of logistics channel cooperation. The government, as the leader of the logistics channel in the national unified market, should aim at improving cooperation benefits, create favorable conditions for enterprises to create a fair and reasonable income distribution environment, release the principal documents in time at the macro level, and actively build a cross-regional cooperation and exchange platform and cooperation risk mitigation mechanism to help cooperative enterprises along the route avoid risks, enhance mutual trust, deepen cooperation and increase profits. This will be conducive to participating in the interests of enterprises, the benefits of logistics channel system and social welfare to achieve a win-win situation. In addition, in the process of the government implementation of reward and punishment measures, it is necessary to provide a timely transformation of the results of cooperation into feedback policies to create 'performance-based rewards'. In the process of logistics channel cooperation, the participants should be supervised, and when passive work is found, the 'punishment measures' should be implemented in time to avoid the loss of income caused by improper subsidies.

**(3) Optimize the cross-regional logistics coordination and interaction mechanism to reduce management costs.** Through the above evolutionary game and simulation analysis, it has been shown that when the proportion of cost and total amount of logistics channel construction are relatively low, the efficiency gains from participating enterprises achieving cooperation equilibrium is highest. In order to increase the economic benefits of this cross-regional

logistics coordination, it is necessary to reduce the corresponding management costs. As far as trans-regional logistics management is concerned, the cooperation mode of domestic logistics enterprises is still in a relatively backward state. The government should guide logistics enterprises to use advanced technology, refer to successful models, strengthen infrastructure construction such as trans-regional transport corridors, and realize division of labor and cooperation and complementary advantages. On the basis of minimizing the management cost, realizing the mutual benefit mechanism with low cost and high income, and forming a trans-regional logistics channel management system with the government as the leading factor, the market as the link and enterprises as the main body, will promote the regional logistics integration process.

(4) **The government should implement reasonable rewards and punishments to mobilize the participation enthusiasm of logistics enterprises.** Through the evolutionary game and simulation analysis of government rewards and punishments, it can be seen that strengthening rewards and punishments under the guidance of the government to attract logistics enterprises to participate in channel construction is conducive to achieving the balance of interests. However, setting the level of government rewards and punishments is not simply a matter of the higher the better. When the government rewards and punishments are too high, the reward and punishment strategy should be moderated, the input cost should be reasonably reduced, and the cost structure should be optimized in a diversified way, so as to prevent the positive guiding role from being impaired by moral hazard, leading to the enterprises' cooperation behavior being suboptimal when driven by economic interests. Cross-regional logistics channel performance requires funding from government revenue, resulting in local financial pressure. Therefore, the government's reward and punishment measures have utility boundaries, which are in line with the differentiated strategies of different levels and different radiation ranges in practice, so as to fully mobilize the positive influence of guiding policies on the implementation of logistics channels, and meet the needs of new economic development and industrial layout for efficient interconnection of logistics capabilities.

## 5.3 Limitations and future research directions

This paper considers the evolutionary game between regional logistics enterprises under market mechanism and government guidance, and explores the dynamic process of enterprises participation in logistics channel strategy selection under government guidance. However, this paper does not consider the government input costs and output benefits when establishing the evolutionary game model, and assumes that the participants of 'government enterprises' are all risk-neutral. In our further work, local governments will be considered into the model to examine the equilibrium under different risk preferences.

## Author Contributions

**Conceptualization:** Junqian Xu.

**Formal analysis:** Guangsheng Zhang, Junqian Xu.

**Investigation:** Yanling Wang.

**Methodology:** Guangsheng Zhang, Yanling Wang.

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
