## [Decision Letter · Decision Letter 0]

17 Jul 2023

PONE-D-23-19733A Study on the Promotion Effect of Government Guidance on the Construction of a National Unified Market Logistics ChannelPLOS ONE

Dear Dr. Xu,

Thank you for submitting your manuscript to PLOS ONE. After careful consideration, we feel that it has merit but does not fully meet PLOS ONE’s publication criteria as it currently stands. Therefore, we invite you to submit a revised version of the manuscript that addresses the points raised during the review process.

Dear authors,

Thank you for choosing PLOS ONE as an outlet for your paper publication. We have received evaluations from experts. The reviewers have highlighted several issues which need to be answered and strengthen the quality of this draft.

Authors should take these comments into account and adjust the manuscript content accordingly.

1. The draft needs to be significantly improved. In addition to the improvements requested by Reviewers, you should expand them in line with the PLOS ONE guidelines and revise the abstract accordingly.

2. Reviewers highlight several issues concerning the introduction, literature review, and conclusion sections. Therefore, Revising these sections to meet the publication criteria of PLOS ONE is suggested.

3. This article requires proofreading to enhance the linguistic quality further. Also, please update the reference list, preferably referencing recent works published in top leading journals.

4. I suggest authors ensure that all the cited articles should be properly listed in the reference section.

We look forward to receiving your revised manuscript.

Kind regards,

Syed Abdul Rehman Khan, PhD

Academic Editor

PLOS ONE

“Acknowledgments

We would like to be grateful to the editors and anonymous referees for their valuable comments. This work was supported by National Social Science Foundation of China, grant number 21BJY227, and the PhD Research Initiation Fund of Shandong University of Management.

Authorship contribution statement

Conceptualization, G.Z. and Y.W.; methodology, G.Z.; software, J.X.; formal analysis, G.Z.; writing-original draft preparation, G.Z.; writing-review and editing, Y.W. All authors have read and agreed to the published version of the manuscript.”

“We would like to be grateful to the editors and anonymous referees for their valuable comments. This work was supported by National Social Science Foundation of China, grant number 21BJY227, and the PhD Research Initiation Fund of Shandong University of Management.”

“Acknowledgments

We would like to be grateful to the editors and anonymous referees for their valuable comments. This work was supported by National Social Science Foundation of China, grant number 21BJY227, and the PhD Research Initiation Fund of Shandong University of Management.

Authorship contribution statement

Conceptualization, G.Z. and Y.W.; methodology, G.Z.; software, J.X.; formal analysis, G.Z.; writing-original draft preparation, G.Z.; writing-review and editing, Y.W. All authors have read and agreed to the published version of the manuscript.”

Reviewers' comments:

Reviewer's Responses to Questions

**Comments to the Author**

1. Is the manuscript technically sound, and do the data support the conclusions?

Reviewer #1: Partly

Reviewer #2: Yes

Reviewer #3: Yes

2. Has the statistical analysis been performed appropriately and rigorously? 

Reviewer #1: Yes

Reviewer #2: Yes

Reviewer #3: Yes

3. Have the authors made all data underlying the findings in their manuscript fully available?

Reviewer #1: Yes

Reviewer #2: Yes

Reviewer #3: Yes

4. Is the manuscript presented in an intelligible fashion and written in standard English?

Reviewer #1: No

Reviewer #2: Yes

Reviewer #3: Yes

5. Review Comments to the Author

Reviewer #1: Authors needs to revise this article under the light of the following comments:

1. What are the research objective? This should be clearly explained in introduction section.

2. Authors needs to discuss the logistics industry and game modeling at a broad level and cite the relevant articles. Following articles can be helpful for authors:

https://doi.org/10.1016/j.jclepro.2020.123127

https://doi.org/10.1108/IJRDM-03-2023-0177

3. Citations are missing. Please recheck all the cited articles and ensure that all cited papers are listed in references.

4. Discussion section should be separate from conclusion section. Authors should include one "Discussion Section" and all the key findings should be presented there with the support of previously published literature.

Reviewer #2: ABSTRACT:

The abstract lacks contextual background and should provide a clear introduction to the significance of logistics channel construction in the national unified market. The study objectives should be expressed properly, and particular findings or conclusions should be given. For greater comprehension, the abstract's language and clarity should be enhanced.

INTRODUCTION:

The introduction establishes a strong basis by highlighting the significance of logistics channel development in the national unified market. However, it would benefit from incorporating relevant citations, clearly stating the research gap and objectives, discussing the significance and practical implications of the research, and improving the overall structure and organization.

LITERATURE REVIEW:

The literature review could be more organized and structured, grouping studies according to key findings or research focus. It should give a more critical review and synthesis of the previous literature, noting gaps or unsolved issues that the present study is attempting to address.

METHODS:

The methodology is well-structured, but further details on the specific simulation setup, assumptions, and criteria used for selecting optimal assignment parameters should be provided. The paper should also consider conducting robustness checks, sensitivity analyses, and discussing the limitations of the numerical experiments.

DISCUSSION AND CONCLUSION:

The conclusion effectively summarizes the research findings and offers valuable management implications. However, it could be strengthened by providing a brief reflection on the significance of the research in the broader context of logistics channel construction and the national unified market.

Reviewer #3: This paper is very interesting and well degined however it could be improved for more clarity.

Suggestions for Improvement:

Introduction: The abstract needs a clear motivation for the research and the significance of studying logistics channel construction in a national unified market. Providing a brief background on the importance of efficient logistics channels in supporting economic development and regional coordination would be beneficial.

Objectives and Research Questions: Clearly state the objectives and research questions addressed in the study. It will help readers understand the purpose and scope of the research more effectively.

Methodology: Provide a concise overview of the evolutionary game theory approach used in the study, highlighting its advantages over traditional game theory in addressing logistics channel construction. Explain the rationale for assuming bounded rationality and how it impacts the model's outcomes.

Results and Discussion: Elaborate on the numerical and Matlab software simulations conducted to explore the impact of various factors on strategic decisions. Present the results clearly and structured, discussing the implications of these findings for logistics enterprises and government policymakers. Additionally, consider presenting any limitations of the simulations and possible areas for future research.

Conclusion: Summarize the essential findings and their implications for logistics channel construction in a national unified market. Highlight the practical significance of the research and provide actionable recommendations for government and logistics enterprises to enhance the coordination and development of logistics channels.

Language and Clarity: Ensure the abstract is written in clear and concise language, avoiding jargon or technical terms that may be unfamiliar to a broad audience. Proofread the abstract to eliminate any grammatical errors or awkward phrasing.

There are a few suggestions to enhance the clarity and structure of the introduction:

Background and Motivation: Start the introduction by providing a concise background on the importance of a unified national market and its benefits for economic development. Clearly state the motivation behind the study, such as the need to overcome market segmentation barriers and promote efficient logistics channels in support of regional coordination and economic integration.

Research Gap and Objectives: Explicitly highlight the research gap in terms of limited consideration of bounded rationality and the role of the government in logistics channel construction. Clearly state the research objectives and questions that will be addressed in the study.

Significance and Contribution: Emphasize the importance and contribution of the study by explaining how it extends existing research. Highlight the practical implications for government policymakers and logistics enterprises in formulating effective support policies, encouraging participation, and achieving win-win outcomes.

Structure of the Paper: Provide an overview of its organization, outlining the subsequent sections and their respective content.

6. PLOS authors have the option to publish the peer review history of their article (what does this mean?). If published, this will include your full peer review and any attached files.

Reviewer #1: No

Reviewer #2: No

Reviewer #3: **Yes: **BIBHAV ADHIKARI

---

## [Author Response · Author response to Decision Letter 0]

16 Sep 2023

Replies to Reviewer #1:

Reviewers' comments 1:

1. What are the research objective? This should be clearly explained in introduction section.

Response: 

 We highly appreciate your valuable comments. To address this issue, We added the purpose of the research in introduction to clarify the main research motivation of this paper. At the same time, we also put forward the research purpose in abstract, so that readers can better understand the content of this paper.

Reviewers' comments 2:

2. Authors needs to discuss the logistics industry and game modeling at a broad level and cite the relevant articles. Following articles can be helpful for authors:

https://doi.org/10.1016/j.jclepro.2020.123127

https://doi.org/10.1108/IJRDM-03-2023-0177

Response: 

We highly appreciate your valuable comments. To address this issue, after reading carefully the important literature recommended by the reviewers，the literature has been incorporated as a citation. At the same time, referring to the modification suggestions, we have added a description of the evolutionary game theory, which makes the logic of the paper clearer.

Reviewers' comments 3:

3. Citations are missing. Please recheck all the cited articles and ensure that all cited papers are listed in references.

Response: 

We highly appreciate your valuable comments. To address this issue, we examined the full text of the paper carefully and reorganized the references.

Reviewers' comments 4:

4. Discussion section should be separate from conclusion section. Authors should include one "Discussion Section" and all the key findings should be presented there with the support of previously published literature.

Response: 

We highly appreciate your valuable comments. To address this issue, we revised the content of conclusion and revelation section, added the previous key literature in conclusion, and more clearly explained the main theoretical contribution and academic value of this paper. 

Replies to Reviewer #2:

Reviewers' comments 1:

ABSTRACT:

The abstract lacks contextual background and should provide a clear introduction to the significance of logistics channel construction in the national unified market. The study objectives should be expressed properly, and particular findings or conclusions should be given. For greater comprehension, the abstract's language and clarity should be enhanced.

Response: 

We highly appreciate your valuable comments. To address this issue, we clearly introduce the significance of the logistics channel construction, and clarify the main research purpose of this paper in abstract. At the same time, we modify and comb the language description of the abstract, so that readers can better understand the content of this paper.

Reviewers' comments 2:

INTRODUCTION:

The introduction establishes a strong basis by highlighting the significance of logistics channel development in the national unified market. However, it would benefit from incorporating relevant citations, clearly stating the research gap and objectives, discussing the significance and practical implications of the research, and improving the overall structure and organization.

Response: 

We highly appreciate your valuable comments. To address this issue, we clarify the important role of the government in the construction of logistics channels in introduction, and put forward the main research purpose and practical significance of this paper, which improves the readability and standardization of this paper.

Reviewers' comments 3:

LITERATURE REVIEW:

The literature review could be more organized and structured, grouping studies according to key findings or research focus. It should give a more critical review and synthesis of the previous literature, noting gaps or unsolved issues that the present study is attempting to address.

Response: 

 We highly appreciate your valuable comments. To address this issue, we have divided the relevant studies into two categories, one is the literature on the construction of logistics channels with government participation, and the other is the literature on the application of game theory in the coordination of logistics channel stakeholders. We point out the differences between this study and the current literature and illustrate the key issues that this paper seeks to address.

Reviewers' comments 4:

METHODS:

The methodology is well-structured, but further details on the specific simulation setup, assumptions, and criteria used for selecting optimal assignment parameters should be provided. The paper should also consider conducting robustness checks, sensitivity analyses, and discussing the limitations of the numerical experiments.

Response: 

We highly appreciate your valuable comments. To address this issue, in the numerical experiment section, we illustrate the principle of setting basic parameters in this paper, point out the value and specific range of the model. At the same time, sensitivity analysis is used to clarify the influence mechanism of initial participation intention, positive cooperation benefit, cost proportion coefficient, penalty coefficient and construction cost under the two modes.

Reviewers' comments 5:

DISCUSSION AND CONCLUSION:

The conclusion effectively summarizes the research findings and offers valuable management implications. However, it could be strengthened by providing a brief reflection on the significance of the research in the broader context of logistics channel construction and the national unified market.

Response: 

We highly appreciate your valuable comments. To address this issue, from a broader background, we propose that the logistics channel can promote the flow of commodity elements in a wider range in management enlightenment, which is conducive to the construction of a unified national market.

Replies to Reviewer #3:

Reviewers' comments 1:

Introduction: The abstract needs a clear motivation for the research and the significance of studying logistics channel construction in a national unified market. Providing a brief background on the importance of efficient logistics channels in supporting economic development and regional coordination would be beneficial.

Response: 

We highly appreciate your valuable comments. To address this issue, in abstract, we clearly introduce the significance of logistics channel construction, clarify the main research purpose of this paper, and introduce the importance and significance of efficient logistics channels in supporting economic development and regional coordination.

Reviewers' comments 2:

Objectives and Research Questions: Clearly state the objectives and research questions addressed in the study. It will help readers understand the purpose and scope of the research more effectively.

Response: 

We highly appreciate your valuable comments. To address this issue, we clearly illustrate the purpose and the research questions involved in introduction, which helps the reader to more effectively understand the purpose and scope of the research.

Reviewers' comments 3:

Methodology: Provide a concise overview of the evolutionary game theory approach used in the study, highlighting its advantages over traditional game theory in addressing logistics channel construction. Explain the rationale for assuming bounded rationality and how it impacts the model's outcomes.

Response: 

We highly appreciate your valuable comments. To address this issue, we clearly explain the difference between the evolutionary game theory and the traditional game theory in introduction, and show that the evolutionary game theory can more truly reflect the diversity and complexity of the logistics channel actors.

Reviewers' comments 4:

Results and Discussion: Elaborate on the numerical and Matlab software simulations conducted to explore the impact of various factors on strategic decisions. Present the results clearly and structured, discussing the implications of these findings for logistics enterprises and government policymakers. Additionally, consider presenting any limitations of the simulations and possible areas for future research.

Response: 

We highly appreciate your valuable comments. To address this issue, we propose model limitations and point out possible areas for future research directions.

Reviewers' comments 5:

Conclusion: Summarize the essential findings and their implications for logistics channel construction in a national unified market. Highlight the practical significance of the research and provide actionable recommendations for government and logistics enterprises to enhance the coordination and development of logistics channels.

Response: 

We highly appreciate your valuable comments. To address this issue, in the part of management enlightenment, based on a broader background, we propose that the construction of logistics channels will help to strengthen the smooth connectivity of the domestic market, promote the active connection between the domestic market and the international market, and thus be conducive to the construction of a unified national market. Finally, we provide operational suggestions for the government and logistics enterprises to strengthen the coordination and development of logistics channels.

Reviewers' comments 6:

Language and Clarity: Ensure the abstract is written in clear and concise language, avoiding jargon or technical terms that may be unfamiliar to a broad audience. Proofread the abstract to eliminate any grammatical errors or awkward phrasing.

Response: 

We highly appreciate your valuable comments. To address this issue, we reproofread the full-text language to make it more clear and concise. At the same time, we also proofread the abstract section again to avoid grammatical errors.

---

## [Decision Letter · Decision Letter 1]

3 Oct 2023

PONE-D-23-19733R1A Study on the Promotion Effect of Government Guidance on the Construction of a National Unified Market Logistics ChannelPLOS ONE

Dear Dr. Xu,

Thank you for submitting your manuscript to PLOS ONE. After careful consideration, we feel that it has merit but does not fully meet PLOS ONE’s publication criteria as it currently stands. Therefore, we invite you to submit a revised version of the manuscript that addresses the points raised during the review process. Please submit your revised manuscript by Nov 17 2023 11:59PM. If you will need more time than this to complete your revisions, please reply to this message or contact the journal office at plosone@plos.org. Please include the following items when submitting your revised manuscript:A rebuttal letter that responds to each point raised by the academic editor and reviewer(s). You should upload this letter as a separate file labeled 'Response to Reviewers'.A marked-up copy of your manuscript that highlights changes made to the original version. You should upload this as a separate file labeled 'Revised Manuscript with Track Changes'.An unmarked version of your revised paper without tracked changes. You should upload this as a separate file labeled 'Manuscript'.

We look forward to receiving your revised manuscript.

Kind regards,

Syed Abdul Rehman Khan, PhD

Academic Editor

PLOS ONE

Reviewers' comments:

Reviewer's Responses to Questions

**Comments to the Author**

1. If the authors have adequately addressed your comments raised in a previous round of review and you feel that this manuscript is now acceptable for publication, you may indicate that here to bypass the “Comments to the Author” section, enter your conflict of interest statement in the “Confidential to Editor” section, and submit your "Accept" recommendation.

Reviewer #1: All comments have been addressed

Reviewer #2: All comments have been addressed

Reviewer #3: All comments have been addressed

2. Is the manuscript technically sound, and do the data support the conclusions?

Reviewer #1: Yes

Reviewer #2: Yes

Reviewer #3: Yes

3. Has the statistical analysis been performed appropriately and rigorously? 

Reviewer #1: Yes

Reviewer #2: Yes

Reviewer #3: Yes

4. Have the authors made all data underlying the findings in their manuscript fully available?

Reviewer #1: Yes

Reviewer #2: Yes

Reviewer #3: Yes

5. Is the manuscript presented in an intelligible fashion and written in standard English?

Reviewer #1: Yes

Reviewer #2: Yes

Reviewer #3: Yes

6. Review Comments to the Author

Reviewer #1: Authors have incorporated all the raised comments.

Reviewer #2: I thank you for your devotion and hard work in enhancing your study. Your diligence in responding to prior reviews and improving the overall quality is commendable.

While your modifications have significantly improved the work, I detected a few grammatical problems throughout the text that may need further attention. These are flaws that, once addressed, will improve the paper's general readability and professionalism.

Some examples of the grammar errors I discovered:

Abstract

1. "and to realize regional economic integration, it is of great value to the implementation of the national unified market strategy."

a. The phrase "it is of great value to the implementation" is a bit awkward. It could be rephrased for clarity, such as "it greatly contributes to the implementation."

2. "clarifies the effect of government participation and support policies by clarifying the role and functions of the government"

a. The repetition of "clarifying" is redundant. It can be improved by saying "by defining the role and functions of the government."

3. "We constructs an evolutionary game model among participating stakeholders, then studies the evolutionary stabiliy strategy"

a. "We constructs" should be corrected to "We construct." It's a subject-verb agreement issue.

b. "studies the evolutionary stabiliy strategy" should be corrected to "study the evolutionary stability strategy."

4. "influences each other, and government participation can change the effects of cooperative income,"

a. The subject-verb agreement is incorrect. It should be "influence each other" to match the plural subject.

5. "the government should provide reasonable guidance to prevent adverse selection of enterprises from hindering the healthy development of logistics channels."

a. The phrase "prevent adverse selection of enterprises from hindering" is somewhat complex. It could be simplified to improve readability, such as "to prevent enterprises from hindering."

Introduction

• The language used is generally clear and academic. However, there are some areas where sentence structure could be simplified for greater clarity and readability. For instance, in the sentence "differences in logistics channel participation behavior under the guidance of market mechanism and government are compared, and the difficulty of realizing and maintaining system stability in the two modes is analyzed," it might be clearer to split this into two sentences.

I recommend proofreading the entire work to ensure it is error-free and polished. Addressing these minor issues will elevate the overall quality of the paper.

Reviewer #3: Well Addressed. I have no further Comments. Best wishes and will be happy to review similar paper in comming future as well.

7. PLOS authors have the option to publish the peer review history of their article (what does this mean?). If published, this will include your full peer review and any attached files.

Reviewer #1: No

Reviewer #2: No

Reviewer #3: **Yes: **BIBHAV ADHIKARI

---

## [Author Response · Author response to Decision Letter 1]

11 Oct 2023

Dear Editors and Reviewers:

We appreciate Editor’s sincere letter and Reviewers' helpful comments concerning our manuscript entitled “A Study on the Promotion Effect of Government Guidance on the Construction of a National Unified Market Logistics Channel” (PONE-D-23-19733). Those comments are very valuable and helpful for revising and improving our paper. We have carefully revised the manuscript on the basis of the comments and suggestions. We highlight the changes in our manuscript with track changes in color.

Replies to Reviewer #2:

Reviewers' comments 1:

Abstract

1. "and to realize regional economic integration, it is of great value to the implementation of the national unified market strategy."

a. The phrase "it is of great value to the implementation" is a bit awkward. It could be rephrased for clarity, such as "it greatly contributes to the implementation."

2. "clarifies the effect of government participation and support policies by clarifying the role and functions of the government"

a. The repetition of "clarifying" is redundant. It can be improved by saying "by defining the role and functions of the government."

3. "We constructs an evolutionary game model among participating stakeholders, then studies the evolutionary stabiliy strategy"

a. "We constructs" should be corrected to "We construct." It's a subject-verb agreement issue.

b. "studies the evolutionary stabiliy strategy" should be corrected to "study the evolutionary stability strategy."

4. "influences each other, and government participation can change the effects of cooperative income,"

a. The subject-verb agreement is incorrect. It should be "influence each other" to match the plural subject.

5. "the government should provide reasonable guidance to prevent adverse selection of enterprises from hindering the healthy development of logistics channels."

a. The phrase "prevent adverse selection of enterprises from hindering" is somewhat complex. It could be simplified to improve readability, such as "to prevent enterprises from hindering."

Response: 

We highly appreciate your valuable comments. To address this issue, we have polished the abstract on the basis of the modification requirements, which improved the readability and professionalism of abstract.

Reviewers' comments 1:

Introduction

The language used is generally clear and academic. However, there are some areas where sentence structure could be simplified for greater clarity and readability. For instance, in the sentence "differences in logistics channel participation behavior under the guidance of market mechanism and government are compared, and the difficulty of realizing and maintaining system stability in the two modes is analyzed," it might be clearer to split this into two sentences.

Response: 

We highly appreciate your valuable comments. To address this issue, We simplified the sentence structure according to the modification requirements, “we compares the differences of the participation behavior of the logistics channel between market mechanism and government guidance, analyzed the difficulty of realizing and maintaining the system stability under these two modes. ”

 In addition, we proofread the entire work, which further improved the overall quality of this paper.

Once again, we would like to stress that we highly appreciate the valuable comments and suggestions. We hope the revised manuscript could eventually satisfy your requirements.

 Sincerely,

Guangsheng Zhang

Shandong Management University,

Jinan, Shandong, 250357, P R China.

---

## [Decision Letter · Decision Letter 2]

23 Oct 2023

A Study on the Promotion Effect of Government Guidance on the Construction of a National Unified Market Logistics Channel

PONE-D-23-19733R2

Dear Dr. Xu,

We’re pleased to inform you that your manuscript has been judged scientifically suitable for publication and will be formally accepted for publication once it meets all outstanding technical requirements.

Kind regards,

Syed Abdul Rehman Khan, PhD

Academic Editor

PLOS ONE

Reviewers' comments:

Reviewer's Responses to Questions

**Comments to the Author**

1. If the authors have adequately addressed your comments raised in a previous round of review and you feel that this manuscript is now acceptable for publication, you may indicate that here to bypass the “Comments to the Author” section, enter your conflict of interest statement in the “Confidential to Editor” section, and submit your "Accept" recommendation.

Reviewer #2: All comments have been addressed

2. Is the manuscript technically sound, and do the data support the conclusions?

Reviewer #2: (No Response)

3. Has the statistical analysis been performed appropriately and rigorously? 

Reviewer #2: Yes

4. Have the authors made all data underlying the findings in their manuscript fully available?

Reviewer #2: Yes

5. Is the manuscript presented in an intelligible fashion and written in standard English?

Reviewer #2: Yes

6. Review Comments to the Author

Reviewer #2: (No Response)

7. PLOS authors have the option to publish the peer review history of their article (what does this mean?). If published, this will include your full peer review and any attached files.

Reviewer #2: No

---

## [Editor Report · Acceptance letter]

3 Nov 2023

PONE-D-23-19733R2 

A study on the promotion effect of government guidance on the construction of a national unified market logistics channel 

Dear Dr. Xu:

I'm pleased to inform you that your manuscript has been deemed suitable for publication in PLOS ONE. Congratulations! Your manuscript is now with our production department. 

Kind regards, 

on behalf of

Dr. Syed Abdul Rehman Khan 

Academic Editor

PLOS ONE